# Unveiling structure-performance relationships from multi-scales in non-fullerene organic photovoltaics

Shuixing Li[1,5], Lingling Zhan[1,5], Nannan Yao[2], Xinxin Xia[3], Zeng Chen[4], Weitao Yang[1], Chengliang He[1], Lijian Zuo ●[1✉], Minmin Shi[1], Haiming Zhu ●[4], Xinhui Lu[3], Fengling Zhang ●[2] & Hongzheng Chen ●[1✉]

Unveiling the correlations among molecular structures, morphological characteristics, macroscopic properties and device performances is crucial for developing better photovoltaic materials and achieving higher efficiencies. To achieve this goal, a comprehensive study is performed based on four state-of-the-art non-fullerene acceptors (NFAs), which allows to systematically examine the above-mentioned correlations from different scales. It's found that extending conjugation of NFA shows positive effects on charge separation promotion and non-radiative loss reduction, while asymmetric terminals can maximize benefits from both terminals. Another molecular optimization is from alkyl chain tuning. The shortened alkyl side chain results in strengthened terminal packing and decreased π-π distance, which contribute high carrier mobility and finally the high charge collection efficiency. With the most-acquired benefits from molecular structure and macroscopic factors, PM6:BTP-S9-based organic photovoltaics (OPVs) exhibit the optimal efficiency of 17.56% (certified: 17.4%) with a high fill factor of 78.44%, representing the best among asymmetric acceptor based OPVs. This work provides insight into the structure-performance relationships, and paves the way toward high-performance OPVs via molecular design.

[1] State Key Laboratory of Silicon Materials, MOE Key Laboratory of Macromolecular Synthesis and Functionalization, Department of Polymer Science and Engineering, Zhejiang University, Hangzhou, P. R. China. [2] Department of Physics, Chemistry and Biology (IFM), Linköping University, Linköping, Sweden. [3] Department of Physics, Chinese University of Hong Kong, New Territories, Hong Kong, P. R. China. [4] Department of Chemistry, Zhejiang University, Hangzhou, P. R. China. [5] These authors contributed equally: Shuixing Li, Lingling Zhan. ✉email: zjuzlj@zju.edu.cn; hzchen@zju.edu.cn

Organic photovoltaics (OPVs) have now entered a period of prosperity with power conversion efficiencies (PCEs) growing up to over 18%, due to the innovations of photoactive materials[1–5]. Especially for nonfullerene acceptors (NFAs), it has been evolved in several generations from early twisted molecules based on diketopyrrolopyrrole (DPP) or perylene diimide (PDI) units[6,7], to efficient acceptor-donor-acceptor (A-D-A) type planar acceptors, typically ITIC[8], then to state-of-the-art A-DA'D-A type Y-series molecules, like Y6[9]. Each evolution of NFAs has been accompanied with the better understanding on the relationships between material designs and morphological or macroscopic factors. For example, ITIC resolved the paradox between suitable phase separation and efficient charge transport by combining bulky D unit with small but strong A terminal, and Y6 enabled multiple charge transport channels and stronger luminescence efficiencies by introducing an electron-deficient core[10–13]. To push OPVs toward a higher efficiency, it's necessary to build the structure-performance correlations among molecular structures, morphological characteristics (e.g., molecular packing and phase separation), macroscopic factors (e.g., carrier mobility, charge recombination, and exciton behaviors), and the device performances. For this purpose, NFAs with delicately tailored molecular structures should be designed to perform the systematic studies. Compared with symmetric NFAs, asymmetric electron acceptors with two different terminals, e.g., $A_1$-D-$A_2$ type molecules, have more possibilities in achieving more functionalities by combining or compromising diverse merits from two terminals[14–16]. For example, our group previously reported asymmetric electron acceptors, BTP-S1 and BTP-S2, by introducing halogenated indandione ($A_1$) and halogenated 3-dicyanomethylene-1-indanone ($A_2$) as two different terminals, leading to high electroluminescence quantum efficiency and efficient charge separation simultaneously[16]. Thus, together with symmetric electron acceptors, asymmetric electron acceptors provide an extra dimension to tailor the molecular structure for the correlation studies.

Device performances of OPVs are composed of three photovoltaic parameters of open-circuit voltage ($V_{oc}$), short-circuit current density ($J_{sc}$), and fill factor (FF), which ultimately determine the overall PCE. Generally, the molecular structures and the subsequent morphological characteristics of active layer are closely related to the macroscopic properties of the thin films, and can further determine the photovoltaic parameters. Currently, single correlation is gradually established. For example, it's recognized that the radiative loss below bandgap and nonradiative loss, which are correlated with energetic disorder and electroluminescence quantum efficiency ($EQE_{EL}$), respectively, are two main channels limiting the open-circuit voltage[17–22]. In principle, a nanoscale phase-separated blend morphology is required to match the exciton diffusion length for efficient charge separation, and dominant π–π stacking orientation of NFAs is also needed for vertical charge transport[23–26]. In NFA systems, the OPVs have been observed to work efficiently at a small exciton separation driving force, and thus carrier dynamics of NFA systems with small driving forces, including hole transfer and exciton lifetime, is also worthy of exploring[27–30]. Nevertheless, comprehensive understanding from molecular structure to device performance is still required to unveil the fundamentals of OPVs and guide the community to higher device performance.

In this work, four NFA systems with two symmetric NFAs of BO-4Cl and BTP-S7 and two asymmetric NFAs of BTP-S8 and BTP-S9 (Fig. 1a), which belong to the state-of-the-art Y-series molecules, were designed and synthesized to systematically study the structure-performance relationships in OPVs. Specifically, the effects of extending conjugation, molecular symmetry, and alkyl chain tuning of NFAs on morphological characteristics, macroscopic properties, and photovoltaic parameters were studied and compared. By pairing with the polymer donor PM6 (Fig. 1b), OPV devices were fabricated and optimized with these four acceptors. Afterwards, detailed analysis in energy loss, blend morphology, and charge carrier dynamics was performed to extract morphological characteristics and macroscopic factors for these four NFA systems. Finally, a comprehensive landscape covering from molecular design, morphological characteristics, macroscopic properties to photovoltaic parameters was presented, and the OPV device based on PM6:BTP-S9 blend exhibited the best performance of 17.56% due to the balance of macroscopic properties as originated from the molecular structure and morphological factors.

## Results

**Synthesis and characterization.** In this work, four NFAs of BO-4Cl, BTP-S7, BTP-S8, and BTP-S9 (see chemical structures in Fig. 1a) were studied and compared, wherein BO-4Cl was a reported molecule and the other three ones were newly developed[31]. Symmetric electron acceptor of BTP-S7 and asymmetric electron acceptors of BTP-S8 and BTP-S9 were all synthesized via one step of Knoevenagel condensation reaction. Although there were by-products of symmetric molecules during the synthesis of BTP-S8 and BTP-S9, these two asymmetric electron acceptors could both be well separated as pure compounds from the mixtures. Detailed information about their synthesis could be found in the Supplementary Information. The differences of these four electron acceptors mainly lie in electron-accepting terminals or alkyl side chains. Two terminals of 2-(5,6-dichloro-3-oxo-2,3-dihydro-1H-inden-1-ylidene)malononitrile (IC-2Cl) and 2-(6,7-difluoro-3-oxo-2,3-dihydro-1H-cyclopenta[b]naphthalen-1-ylidene)malononitrile (NC-2F) were selected with distinct features in conjugation lengths and halogen types, which might endow different features to relevant electron acceptors[32–34]. Alkyl side chains tuning between BTP-S8 and BTP-S9 might also affect the molecular packing, thus leading to different properties[31,35,36]. As results, effects of terminal groups, molecular symmetry, and alkyl side chains on morphological characteristics, macroscopic properties of active layers, and optoelectronic properties of OPVs could be systematically studied to derive the molecular structure-performance relationships.

To verify the effects of tuning terminals or alkyl side chains on the molecular geometries and electrostatic potentials (ESPs), density functional theory (DFT) calculations at B3LYP/6-31 G were performed and the results were presented in Supplementary Fig. 1. It was found that alkyl chains on the pyrrole rings would be more concentrated in the centers for molecules with one or two NC-2F terminals, which might be caused by the steric effect from bulky NC-2F. Consequently, ESPs of alkyl chains on the pyrrole rings would not affect ESPs of terminals in BTP-S7, BTP-S8, and BTP-S9, while not in BO-4Cl. Of course, all four electron acceptors demonstrated positive ESPs, which was beneficial for charge separation[37].

Normally, terminal tuning will affect the energy levels. To compare the energy levels of these four electron acceptors, cyclic voltammetry (CV) method was firstly applied (Fig. 1d and Supplementary Fig. 2). As expected, BTP-S7 owned the highest-lying energy levels of −4.08 eV and −5.52 eV, while BO-4Cl owned the lowest-lying energy levels of −4.10 eV and −5.69 eV. As for BTP-S8 and BTP-S9, they possessed the same lowest unoccupied molecular orbital (LUMO) level of −4.09 eV and the same highest occupied molecular orbital (HOMO) level of −5.63 eV. The above results were comprehensible since IC-2Cl had stronger electron-withdrawing ability than NC-2F. We could also learn that alkyl side chains tuning had

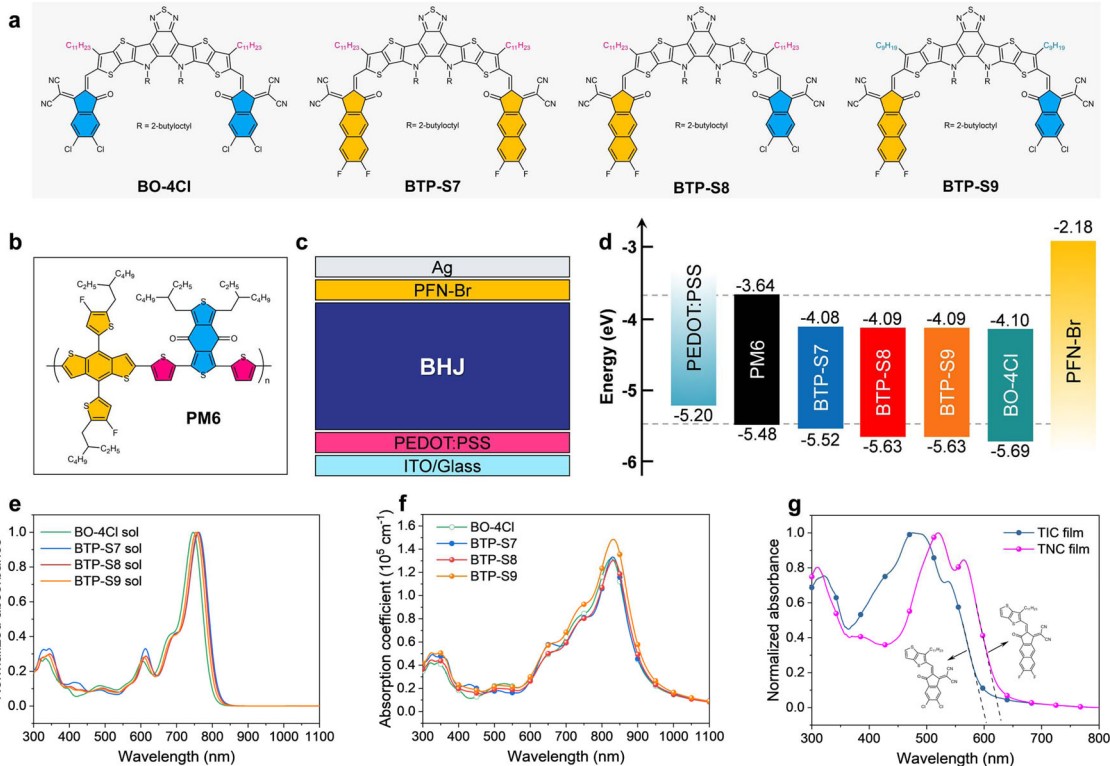

**Fig. 1 Molecular structure, optical, and electrochemical properties. a** Chemical structures of BO-4Cl, BTP-S7, BTP-S8, and BTP-S9. **b** Chemical structure of PM6. **c** Diagram of device structure. **d** Schematic energy level alignment of the studied materials. **e** Normalized absorption spectra of BO-4Cl, BTP-S7, BTP-S8, and BTP-S9 in chloroform solution. **f** Absorption coefficients of BO-4Cl, BTP-S7, BTP-S8, and BTP-S9 thin films. **g** Normalized absorption spectra of TIC and TNC thin films.

little influence on the energy levels. To further confirm the relationships of energy levels, we also calculated the LUMO and HOMO levels via DFT method (Supplementary Fig. 1) and found that the variation tendencies were the same as those from CV method.

For Y-series molecules, a significant advantage is the ordered molecular packing or energetic states, both of which may have an effect on the absorption profiles[38–40]. The absorption spectra of the four electron acceptors in chloroform (CF) solutions and thin films were measured and compared (Fig. 1e-f and Supplementary Fig. 3). In chloroform solutions, BTP-S7 with NC-2F shows more redshifted absorption than BO-4Cl with IC-2Cl. Considering IC-2Cl has stronger electron-withdrawing ability to induce larger intramolecular charge transfer (ICT) effect than NC-2F, there could be more severe aggregation tendency of BTP-S7 even in solution, which was also confirmed by comparing the absorption profiles of two molecular fragments of TIC and TNC composed of thiophthene and relevant terminals (Fig. 1g and Supplementary Fig. 4). For two asymmetric NFAs, BTP-S8, and BTP-S7 in CF solution presented nearly overlapped absorption, while BTP-S9 solution demonstrated a bit blue-shifted absorption. These results indicated that terminal or alkyl side chain tuning had led to different molecular packing behaviors in solution. In thin films, these four NFAs exhibit similar absorption profiles. The BO-4Cl, BTP-S7 and BTP-S8 showed similar absorption coefficients of $\sim1.3\times10^5\,\mathrm{cm}^{-1}$ at the highest peaks, while a higher coefficient of $\sim1.5\times10^5\,\mathrm{cm}^{-1}$ was observed for BTP-S9 at the highest peak. Notably, the similar absorption profiles provide a unique platform to study the individual role of the molecular structure variations on carrier dynamics without entangling with the optical absorbing effect.

**Photovoltaic properties**. To check the photovoltaic properties of OPVs based on four NFAs, PM6 was selected as the polymer donor (Fig. 1b), and a conventional device structure of indium tin oxide (ITO)/PEDOT:PSS/active layer/PFN-Br/Ag (Fig. 1c) was applied to fabricate the devices, wherein PEDOT:PSS is poly(3,4-ethylene-dioxythiophene) poly(styrene sulfonate) and PFN-Br is poly[(9,9-bis(3′-((N,N-dimethyl)-N-ethylammonium)-propyl)-2,7-fluorene)-alt-2,7-(9,9-dioctylfluorene)]. The optimization process can be found in Supplementary Table 1–3. The optimal conditions for BTP-S7 and BTP-S8 were identical in the donor/acceptor (D/A) weight ratio (1:1) and additive (0.5% 1-chloronaphthalene, CN), except the annealing temperature (80 °C for BTP-S7 and 100 °C for BTP-S8). And, the optimal conditions for BO-4Cl and BTP-S9 were identical in the D/A weight ratio (1:1.2) and additive (0.25% 1,8-diiodooctane, DIO), except the annealing temperature (100 °C for BO-4Cl and 80 °C for BTP-S9). The J–V curves of optimal OPVs based on four NFAs were displayed in Fig. 2a, and the relevant photovoltaic parameters were summarized in Table 1.

Obvious contrasts in all the three photovoltaic parameters were observed. For $V_{oc}$, it gradually increased in the order of PM6:BO-4Cl, PM6:BTP-S9, PM6:BTP-S8, and PM6:BTP-S7-based devices, conforming to the elevating trend in the LUMO levels (Supplementary Fig. 5). For $J_{sc}$, PM6:BTP-S8-based OPVs exhibited the highest values, PM6:BTP-S7-based OPVs exhibited the lowest values, while PM6:BO-4Cl and PM6:BTP-S9-based OPVs lied in between. For FF, PM6:BTP-S9-based OPVs showed the highest FF of 78.44%, while the other three OPVs presented similar FF values around 75%. Consequently, OPVs based on two symmetric electron acceptors could realize PCEs of 17.21% (BO-4Cl) and 16.76% (BTP-S7), while those based on two asymmetric electron acceptors could achieve higher PCEs of 17.33% (BTP-S8) and 17.56% (BTP-S9). A certified efficiency of 17.4% was achieved for PM6:BTP-S9-based OPVs from

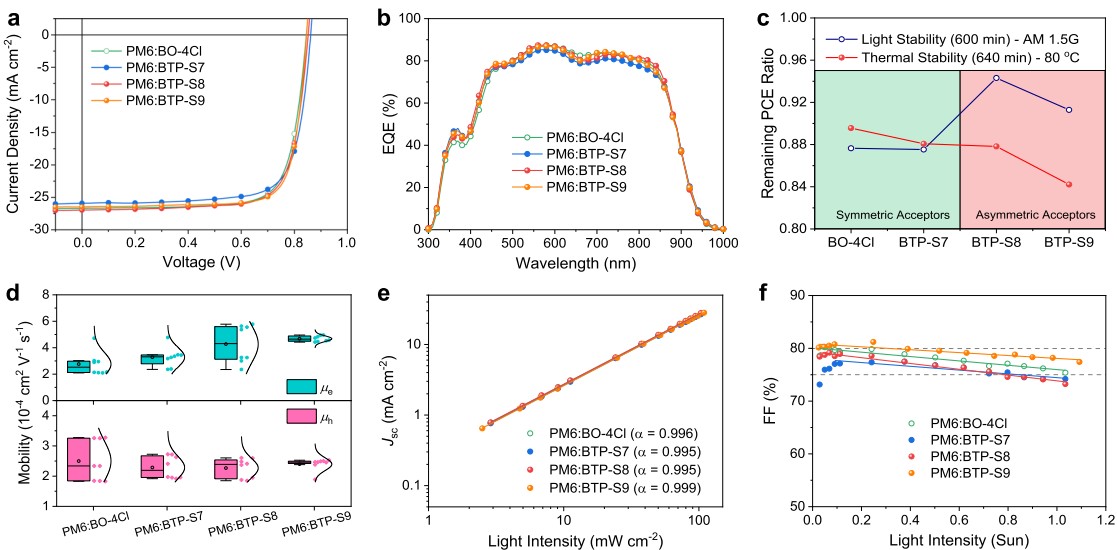

**Fig. 2 Photovoltaic performance, charge transport, and recombination. a** $J$–$V$ curves of OPVs based on PM6:BO-4Cl, PM6:BTP-S7, PM6:BTP-S8, and PM6:BTP-S9 blends. **b** EQE curves of the relevant OPVs. **c** Light stability and thermal stability comparisons of relevant OPVs. **d** Hole mobility and electron mobility comparisons of relevant blends (error bar is defined as the standard deviation, which is calculated from the statistics results of eight devices). **e** The dependence of $J_{sc}$ on light intensity ($P_{light}$) of relevant OPVs. **f** The dependence of FF on $P_{light}$ of relevant OPVs.

National Institute of Metrology (NIM), China (Supplementary Fig. 6). Notably, the PCE of 17.56% was the highest efficiency reported to date for binary OPVs based on asymmetric electron acceptors (Supplementary Fig. 7).

External quantum efficiency (EQE) measurements were performed to examine the properties of photocurrent generation, and the relevant results were illustrated in Fig. 2b. All four types of devices exhibited broad and high EQE values between 450–850 nm. The differences in EQE spectra mainly located in 700–900 nm, which were originated from the absorption of electron acceptors. A detailed comparison revealed that more sharp and higher EQE curve was observed in the range of absorption edges for PM6:BTP-S8-based devices (Supplementary Fig. 8). The calculated photocurrents from the EQE curves were found to be 26.39 mA cm$^{-2}$ for PM6:BO-4Cl-based device, 25.87 mA cm$^{-2}$ for PM6:BTP-S7-based device, 26.56 mA cm$^{-2}$ for PM6:BTP-S8-based device and 26.42 mA cm$^{-2}$ for PM6:BTP-S9-based device, consistent with the $J_{sc}$ values from the $J$–$V$ curves with negligible errors.

Light stability and thermal stability of OPVs based on the four NFAs were examined and compared (Fig. 2c and Supplementary Fig. 9). After light soaking for 600 min or thermal treatment for 640 min, all OPVs could maintain over 84% efficiencies.

**Charge transport and charge recombination**. To better understand the variations in device parameters of these OPVs, we firstly measured the charge transport properties, including hole and electron mobilities, via space-charge limited current (SCLC) method. The results were illustrated in Fig. 2d and Supplementary Figs. 10–11. For neat acceptors, they showed similar electron mobilities in the order of magnitude of $10^{-4}$ cm$^2$ V$^{-1}$ s$^{-1}$ (Supplementary Fig. 10). It was found that the devices based on the four PM6:NFA blends demonstrated similar hole mobilities with the values around $2 \times 10^{-4}$ cm$^2$ V$^{-1}$ s$^{-1}$. However, the electron mobility gradually increased from $2.76 \times 10^{-4}$ cm$^2$ V$^{-1}$ s$^{-1}$ for PM6:BO-4Cl-based devices, $3.28 \times 10^{-4}$ cm$^2$ V$^{-1}$ s$^{-1}$ for PM6:BTP-S7-based devices, to $4.27 \times 10^{-4}$ cm$^2$ V$^{-1}$ s$^{-1}$ for PM6:BTP-S8-based devices, $4.68 \times 10^{-4}$ cm$^2$ V$^{-1}$ s$^{-1}$ for PM6:BTP-S9-based devices.

Charge recombination was detected by measuring the $J$–$V$ curves under various light intensities ($P_{light}$)[41]. The relationship between $V_{oc}$ and $P_{light}$ can be described as $V_{oc} \propto nkT/q\ln(P_{light})$, in which $k$ is the Boltzmann constant, $T$ is the Kelvin temperature, and q is the elementary charge. From the $V_{oc}$ − $P_{light}$ relationship, we could judge whether bimolecular recombination (if n close to 1) or trap-assisted recombination (if n close to 2) was the dominant charge recombination form. As shown in Supplementary Fig. 12, all four types of OPVs presented the same situation that bimolecular recombination was dominant and trap-assisted recombination was well hindered. Degree of bimolecular recombination at short-circuit condition could be reflected in the $J_{sc}$ − $P_{light}$ relationship, which could be described as $J_{sc} \propto P_{light}^{\alpha}$. The more closer to 1 of α value is, the less bimolecular recombination it is. The α values were calculated as 0.996 for PM6:BO-4Cl-based OPVs, 0.995 for both PM6:BTP-S7 and PM6:BTP-S8-based OPVs, and 0.999 for PM6:BTP-S9-based OPVs (Fig. 2e), representing similar situations again. Further, the FF values under various $P_{light}$ were also compared among the four OPVs (Fig. 2f), and PM6:BTP-S9-based OPVs could maintain minimal FF of around 78% and maximal FF of around 81% in the range of 0.02 ~1 sun, both of which were higher than those of other three OPVs, showing that PM6:BTP-S9-based OPVs have reached the least nonideal charge recombination.

**Energy loss analysis**. According to Shockley-Queisser (SQ) limit, energy loss of a device contains three parts, as shown in the following formula[42]:

$$E_{loss} = E_g - qV_{oc} = (E_g - qV_{oc}^{SQ}) + (qV_{oc}^{SQ} - qV_{oc}^{rad}) + (qV_{oc}^{rad} - qV_{oc})$$
$$= (E_g - qV_{oc}^{SQ}) + q\Delta V_{oc}^{rad,below\,gap} + q\Delta V_{oc}^{non-rad} = \Delta E_1 + \Delta E_2 + \Delta E_3$$

(1)

Where $\Delta E_1$ is the radiative loss above bandgap, $\Delta E_2$ is the radiative loss below bandgap, and $\Delta E_3$ is the nonradiative loss. By measuring the electroluminescence quantum efficiency (EQE$_{EL}$) of a device, the nonradiative loss can be easily calculated through the following formula:

$$\Delta E_3 = q\Delta V_{oc}^{non-rad} = -kT\ln(EQE_{EL})$$

(2)

**Table 1 Photovoltaic parameters of the OPVs based on PM6:NFA blends.**

| Blend[a] | $V_{oc}$ (V) | $J_{sc}$ (mA cm$^{-2}$) | PCE (%)[b] | FF (%) | $J_{cal.}$ (mA cm$^{-2}$)[c] | $\mu_h$ (10$^{-4}$ cm$^2$ V$^{-1}$ s$^{-1}$)[d] | $\mu_e$ (10$^{-4}$ cm$^2$ V$^{-1}$ s$^{-1}$)[e] |
|---|---|---|---|---|---|---|---|
| PM6:BO-4Cl | 0.845 (0.846 ± 0.002) | 26.70 (26.62 ± 0.13) | 17.21 (17.02 ± 0.15) | 75.83 (75.20 ± 0.53) | 26.39 | 2.50 ± 0.62 | 2.76 ± 0.83 |
| PM6:BTP-S7 | 0.861 (0.859 ± 0.003) | 25.92 (26.00 ± 0.12) | 16.76 (16.51 ± 0.16) | 75.09 (73.62 ± 0.59) | 25.87 | 2.29 ± 0.35 | 3.28 ± 0.70 |
| PM6:BTP-S8 | 0.852 (0.852 ± 0.002) | 26.96 (26.66 ± 0.36) | 17.33 (17.05 ± 0.11) | 75.45 (74.99 ± 1.05) | 26.56 | 2.27 ± 0.30 | 4.27 ± 1.34 |
| PM6:BTP-S9 | 0.846 (0.848 ± 0.003) | 26.47 (26.48 ± 0.14) | 17.56 (17.42 ± 0.07) | 78.44 (77.72 ± 0.46) | 26.42 | 2.40 ± 0.20 | 4.68 ± 0.20 |

[a]Active area: 9.25 mm$^2$, measured with a mask (area: 5.979 mm$^2$, certified).
[b]Values in the parentheses are the average PCEs based on ten devices.
[c]Calculated current densities from the corresponding EQE curves.
[d]Hole mobilities measured via the SCLC method.
[e]Electron mobilities measured via the SCLC method.

Where $k$ is the Boltzmann constant, $T$ is the temperature in Kelvin. Relevant characterizations to calculate the energy loss can be found in Fig. 3 and Supplementary Fig. 13, and the results of detailed energy losses of OPVs based on four blends were summarized in Table 2.

For $\Delta E_1$, it's unavoidable and all four types of OPVs showed the similar values of $\Delta E_1$ due to their similar band gaps. For $\Delta E_2$, BTP-S7 and BTP-S8-based OPVs had smaller $\Delta E_2$ than BO-4Cl and BTP-S9-based OPVs, which is related with their energetic disorder at the tail-state absorption. The degree of energetic disorder can be quantified with the Urbach energy ($E_U$). The relationship between tail-state absorption ($\alpha(E)$) and $E_U$ follows the Urbach rule described by the following formula[43]:

$$\alpha(E) = \alpha_0 e^{\frac{(E-E_0)}{E_U}} \qquad (3)$$

Where, $\alpha_0$ and $E_0$ are two constants and $E$ is the photon energy. Thus, the smaller $E_U$ means the lower energetic disorder. By measuring the high-resolution Fourier transform photocurrent spectroscopy EQE spectra (FTPS-EQE) or absorption coefficients, we were able to derive $E_U$ through exponential fitting. The Urbach energy values of OPV devices based on the four NFAs exhibit negligible differences, but get decreased compared to those of pristine NFA films as shown in Supplementary Fig. 14. The values of $E_U$ were calculated as 21.37, 21.27, 20.79, and 21.39 meV for devices based on PM6:BO-4Cl, PM6:BTP-S7, PM6:BTP-S8, and PM6:BTP-S9, respectively. BTP-S7 and BTP-S8-based devices indeed possessed smaller $E_U$ values and less energetic disorder than BO-4Cl and BTP-S9-based devices, even though polymer donor PM6 was introduced. To the best of our knowledge, the $E_U$ value of 20.79 meV for PM6:BTP-S8-based device is the lowest in highly efficient OPVs. Therefore, lowering energetic disorder is beneficial for reducing radiative loss below bandgap. For $\Delta E_3$, BTP-S7-based OPVs possessed the lowest $\Delta E_3$, BO-4Cl-based OPVs owned the highest $\Delta E_3$, while BTP-S8 and BTP-S9-based OPVs had the in-between $\Delta E_3$. Finally, the relationships in total energy losses were similar to the relationships in $\Delta E_3$, representing that nonradiative loss was still the main factor limiting the energy loss.

**Film morphology**. Blend morphology is one of the key factors in affecting device performances. Atomic force microscopy (AFM) technology was firstly utilized to study the upper surface topographies. For a preferred blend morphology, bicontinuous interpenetrating networks and nanoscale phase separations are required. From the AFM images (Fig. 4a and Supplementary Fig. 15), the above requirements could be fulfilled for all four blends, and uniform surfaces with nanofiber-like feature were presented. However, different morphology features could also be observed. Comparing PM6:BO-4Cl and PM6:BTP-S7 blends, more slender and dense fibers could be observed in PM6:BTP-S7 blend, possibly due to the molecular aggregation behavior from NC-2F. Although BTP-S8 and BTP-S9 had the same two terminals, the morphology of PM6:BTP-S8 blend resembled to that of PM6:BTP-S7 with more slender and dense fibers, while more obvious phase separation behavior was presented in PM6:BTP-S9 blend, indicating the length of the alkyl side chain could affect the phase separation behavior.

To further study the crystallinity, molecular packing and orientation in films, grazing-incidence wide-angle X-ray scattering (GIWAXS) characterization was performed. For neat acceptor films (Fig. 4c and Supplementary Fig. 16), a dominant π–π stacking peak at q = 1.75 Å$^{-1}$ ($d$ = 3.59 Å) was presented for BTP-S7, BTP-S8, and BTP-S9 in the out-of-plane (OOP) direction, but at q = 1.77 Å$^{-1}$ ($d$ = 3.55 Å) for BO-4Cl. In the in-plane (IP) direction, a lamellar peak at q = 0.39 Å$^{-1}$ ($d$ = 16.10 Å) was

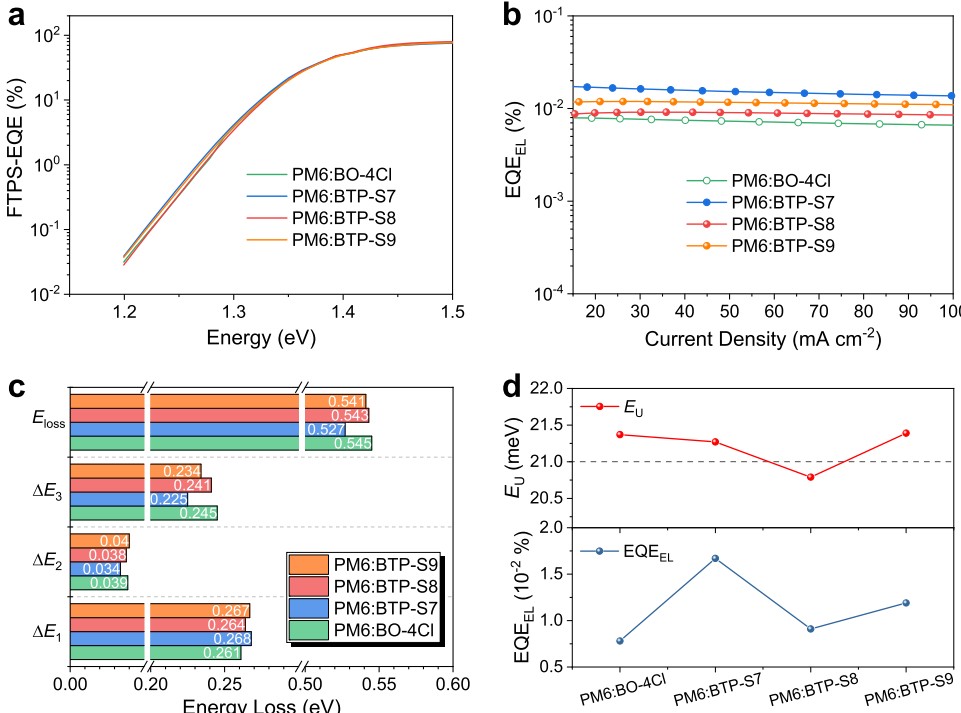

**Fig. 3 Energy loss analysis. a** FTPS-EQE curves of OPVs based on PM6:BO-4Cl, PM6:BTP-S7, PM6:BTP-S8, and PM6:BTP-S9 blends. **b** $EQE_{EL}$ of OPVs at various injected current densities. **c** Comparison of radiative and nonradiative energy losses. **d** Variations of $E_U$ and $EQE_{EL}$ in OPVs based on PM6:NFA blends.

**Table 2 Detailed energy losses of OPVs based on PM6:BO-4Cl, PM6:BTP-S7, PM6:BTP-S8, and PM6:BTP-S9 blends.**

| Devices | $E_g$ (eV) | $qV_{oc}^{SQ}$ (eV) | $qV_{oc}^{rad}$ (eV) | $\Delta E_1$ (eV) | $\Delta E_2$ (eV) | $\Delta E_3$ (eV) | $E_{loss}$ (eV) | $EQE_{EL}$ (%) |
|---|---|---|---|---|---|---|---|---|
| PM6:BO-4Cl | 1.376 | 1.114 | 1.076 | 0.261 | 0.039 | 0.245 | 0.545 | $0.78 \times 10^{-2}$ |
| PM6:BTP-S7 | 1.374 | 1.106 | 1.072 | 0.268 | 0.034 | 0.225 | 0.527 | $1.67 \times 10^{-2}$ |
| PM6:BTP-S8 | 1.379 | 1.114 | 1.077 | 0.264 | 0.038 | 0.241 | 0.543 | $0.91 \times 10^{-2}$ |
| PM6:BTP-S9 | 1.382 | 1.114 | 1.075 | 0.267 | 0.040 | 0.234 | 0.541 | $1.19 \times 10^{-2}$ |

observed for BO-4Cl, BTP-S8 and BTP-S9, but at q = 0.37 Å$^{-1}$ (d = 16.97 Å) for BTP-S7. All four electron acceptors showed face-on orientations. For PM6:NFA blends (Fig. 4d), all four blends showed a lamellar peak at q = 0.30 Å$^{-1}$ (d = 20.93 Å) from polymer donor PM6 in the OOP direction. However, the lamellar peaks of acceptors in the IP direction also moved to q = 0.30 Å$^{-1}$, under the influences of polymer donor PM6, indicating strong influence of PM6 on the molecular packing of NFAs. In addition, a backbone ordering peak at q = 0.40 Å$^{-1}$ (d = 15.70 Å) in the IP direction appeared for the PM6:BTP-S9 blend. As for the (010) π–π stacking peak, it was located at q = 1.73 Å$^{-1}$ (d = 3.63 Å) for both PM6:BTP-S7 and PM6:BTP-S8 blends, but at q = 1.74 Å$^{-1}$ (d = 3.61 Å) for PM6:BO-4Cl blend and q = 1.75 Å$^{-1}$ (d = 3.59 Å) for PM6:BTP-S9 blend. Above distinct features of molecular packing from the acceptors in the blends might also reveal the molecular packing information in the single crystals of relevant NFAs, as demonstrated in previous works[39,44]. Comparing with neat acceptor films, only PM6:BTP-S9 blend could maintain the position of π–π stacking peak as that of neat acceptor, after introducing polymer donor PM6, while the other three blends would all be affected. Thus, the tightest π–π stacking was achieved in PM6:BTP-S9 blend, and the retention of the pristine NFA crystalline structure indicates more pure phase in blend. Besides, higher crystallinity was also observed in PM6:BTP-S9 blend, than those in PM6:BTP-S7 and PM6:BTP-S8 blends. Thereby, the tighter molecular packing and higher crystallinity enabled PM6:BTP-S9

blend possessing the highest electron mobility among the four blends as shown above.

Different molecular aggregation behaviors presented in AFM images possibly led to different domain sizes. To quantify the domain sizes, grazing-incidence small-angle X-ray scattering (GISAXS) characterization was also performed. The results were displayed in Fig. 4e and Supplementary Fig. 17. It was found that the IP domain sizes of pure acceptor phases were 22, 30, 35 and 53 nm in PM6:BO-4Cl, PM6:BTP-S7, PM6:BTP-S8, and PM6:BTP-S9 blends, respectively. The results of domain sizes agreed well with the morphology observed in AFM images.

**Hole transfer dynamics analysis.** Photoinduced charge carrier dynamics is also crucial in determining the OPV efficiency. First, we used time-resolved photoluminescence spectroscopy (TRPL) to measure the lifetimes of neat NFA and PM6:NFA blend films (Supplementary Fig. 18). We found that all NFA films demonstrated a long exciton lifetime of ~1 ns (0.88 ns for BO-4Cl, 0.86 ns for BTP-S7, 0.92 ns for BTP-S8, 0.83 ns for BTP-S9). For the blend films, all four blend films showed a fast PL decay limited by instrument response, suggesting fast interfacial charge transfer. Therefore, we turn to ultrafast transient absorption spectroscopy (TAS) to examine the charge carrier behaviors in the blend films.

Among all four types of OPVs, LUMO offsets were all large enough (>0.4 eV) for electron transfer, while HOMO offsets were

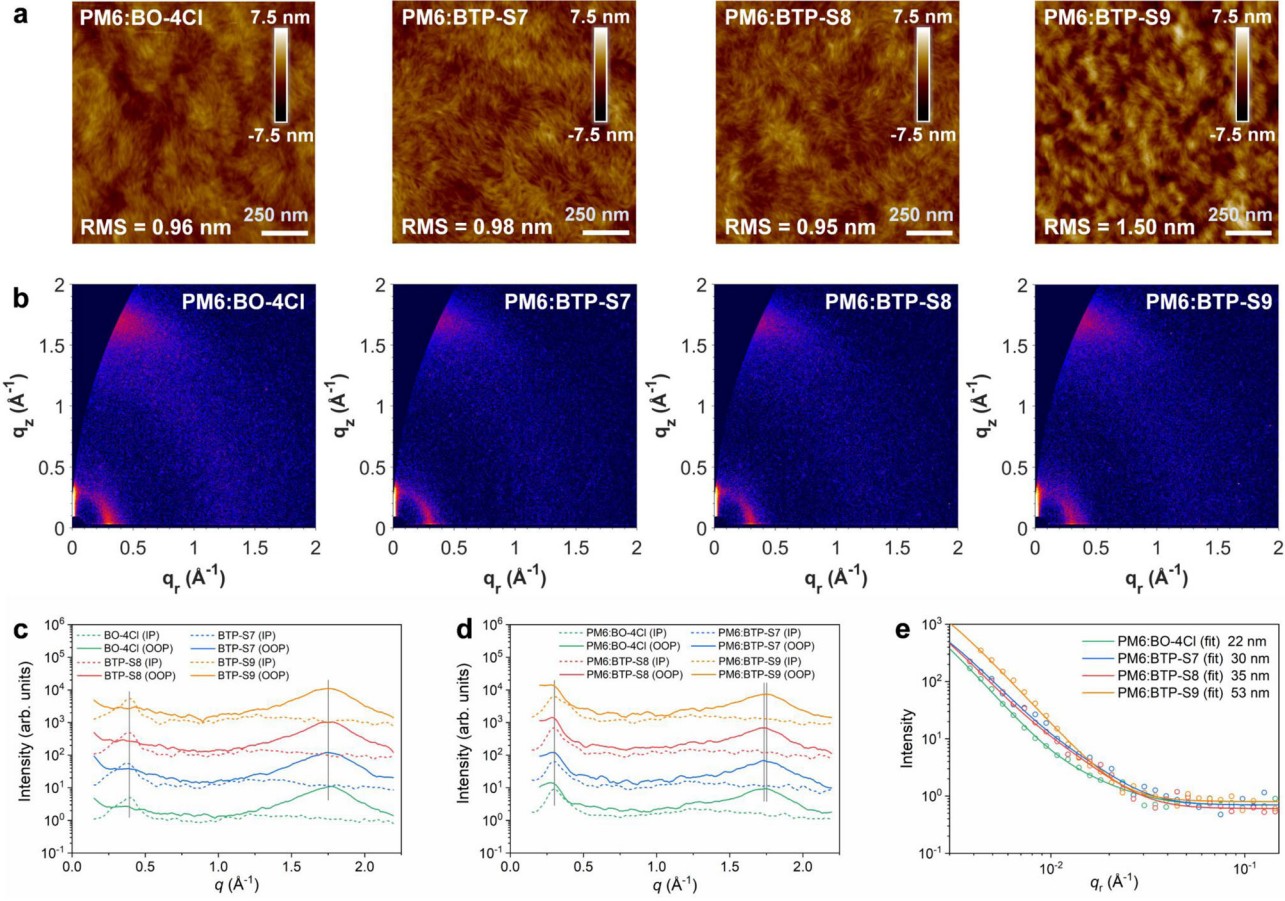

**Fig. 4 Morphological characterization for films. a** AFM height images of PM6:BO-4Cl, PM6:BTP-S7, PM6:BTP-S8, and PM6:BTP-S9 blends. **b** 2D GIWAXS images of relevant blends. **c** 1D intensity profiles of BO-4Cl, BTP-S7, BTP-S8, and BTP-S9 neat films. **d** 1D intensity profiles of relevant blends. **e** The intensity profiles along $q_r$ axis of relevant blends, extracted from 2D GISAXS data.

all less than 0.3 eV, thus we here mainly focused on the hole transfer dynamics. Since the absorption of PM6 and NFAs are well separated, 750 nm excitation was chosen to selectively excite the NFAs. The TA spectra of neat BTP-S9 and PM6:BTP-S9 blend are shown in Fig. 5 and those of other NFAs in Supplementary Fig. 19. For neat BTP-S9 (Fig. 5a), three bleach peaks located at ~670, ~760, and ~850 nm were observed, conforming to its ground state absorption and stimulated emission. The decay of these TA peaks is due to the intrinsic excited state relaxation of BTP-S9. For PM6:BTP-S9 blend (Fig. 5b), the bleach peaks of BTP-S9 at ~850 nm also appears right after photoexcitation. With the increase of decay time, TA signal of BTP-S9 decays quickly. On the other hand, the new TA signals between 600 and 650 nm emerges, which matches well with ground state bleach of PM6. This indicates photoinduced hole transfer process from BTP-S9 to PM6. Similar TA spectral evolutions are observed in other PM6:NFA blends (Supplementary Fig. 19). We choose TA bleach kinetics of PM6 where NFAs show no TA signal to represent hole transfer kinetics in four blends (Fig. 5c).

The hole transfer process can be best described by two time constants of $\tau_1$ and $\tau_2$, in which $\tau_1$ represents the ultrafast exciton dissociation time at the interfaces and $\tau_2$ represents the slower exciton diffusion assisted interfacial hole transfer process[45]. As displayed in Fig. 5d, ultrafast hole transfer process were observed in all blends ($\tau_1 = 0.152 \pm 0.015$ ps for PM6:BO-4Cl, $0.144 \pm 0.014$ ps for PM6:BTP-S7, $0.133 \pm 0.013$ ps for PM6:BTP-S8, $0.160 \pm 0.016$ ps for PM6:BTP-S9). Although PM6:BTP-S7 blend

had smaller HOMO offset than PM6:BO-4Cl, faster hole transfer speed was observed in PM6:BTP-S7 blend, indicating NC-2F with more delocalized electron cloud distribution (Supplementary Fig. 1) was beneficial for faster charge transfer than IC-2Cl. In addition, PM6:BTP-S8 and PM6:BTP-S9 blends had the same HOMO offset, but PM6:BTP-S8 blend could show faster hole transfer speed at the interfaces than PM6:BTP-S9, which might be related with more slender and dense fiber-like morphology in PM6:BTP-S8 blend. $\tau_2$ was much longer than $\tau_1$ ($\tau_2 = 5.45 \pm 0.55$ ps for PM6:BO-4Cl, $3.62 \pm 0.36$ ps for PM6:BTP-S7, $4.70 \pm 0.47$ ps for PM6:BTP-S8, $3.97 \pm 0.40$ ps for PM6:BTP-S9), which were related with charge transport and phase separation properties. Thus, by combining the values of $\tau_1$ and $\tau_2$ and their relevant proportions, we could acquire the total average times for hole transfer process ($\tau_h$) as $1.69 \pm 0.17$ ps, $0.89 \pm 0.09$ ps, $1.31 \pm 0.13$ ps, and $1.16 \pm 0.12$ ps for PM6:BO-4Cl, PM6:BTP-S7, PM6:BTP-S8, and PM6:BTP-S9 blends, respectively.

**Correlation analysis.** In this section, we will first analyze the relationship between molecular structures and morphological characteristics, then try to correlate molecular structures with macroscopic factors, and finally discuss the echoing between macroscopic factors and device performances. The information of molecular structures, morphological characteristics, macroscopic factors, and device performances is summarized in Fig. 6 for correlation analysis.

These molecular structure tunings, concentrated on the terminal sides, are expected to influence the molecular packing

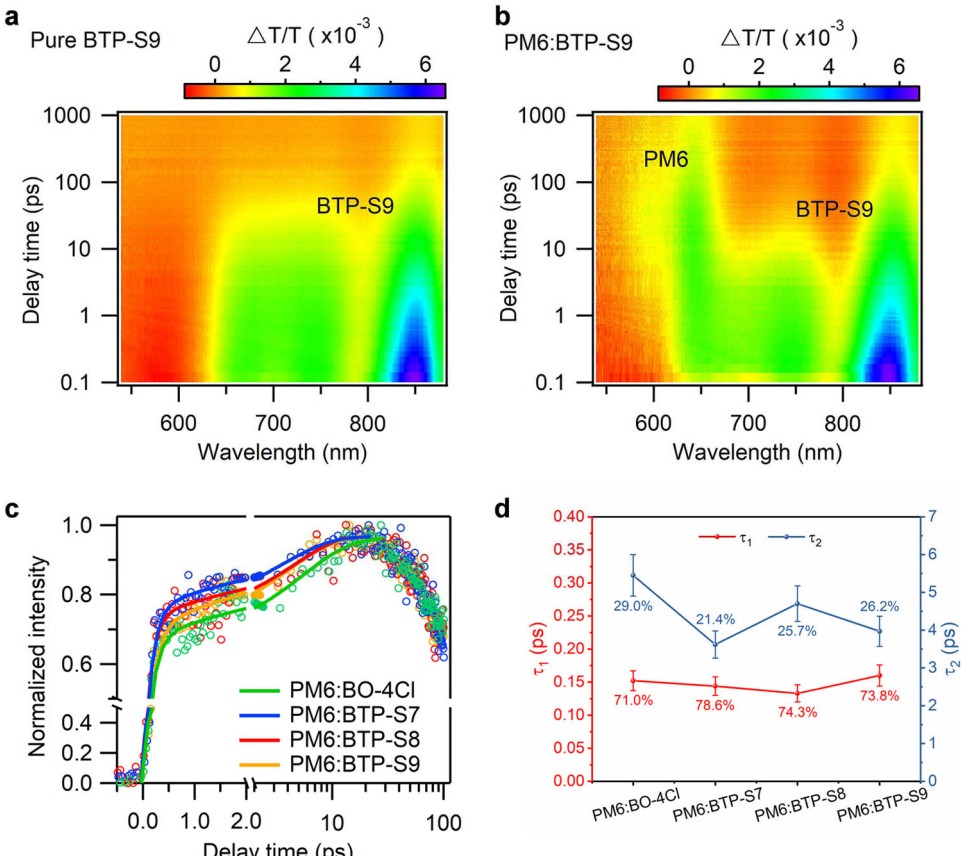

**Fig. 5 Hole transfer dynamics.** Color plot of TA spectra of **a** neat BTP-S9 film and **b** PM6:BTP-S9 blend film under 750 nm excitation. **c** Hole transfer kinetics in four blend films. **d** Comparisons of $\tau_1$ and $\tau_2$ of different blends.

behaviors or aggregation tendencies, thus forming strong correlations with morphological characteristics. In our case, the four NFAs involve three molecular structural factors, i.e., extended conjugation from BO-4Cl to BTP-S7, molecular asymmetry from BTP-S7 to BTP-S8, and alkyl chain length from BTP-S8 to BTP-S9 (see Fig. 6a). The surface roughness (RMS), π–π stacking distance ($L_{(010)}$) and domain size of pure acceptor phase (in the D:A blends) are selected as the morphological characteristics for comparison (Fig. 6b). With long alkyl side chain (C11) fixed, conjugation extension from BO-4Cl to BTP-S7 may lead to increased RMS and domain size of pure acceptor phase, while molecular asymmetry from BTP-S7 to BTP-S8 further enlarges the domain size of pure acceptor phase. Notably, shortening alkyl side chain (C9) from BTP-S8 to BTP-S9 increases RMS and domain size of the pure acceptor phase, indicating terminal packing may be strengthened for BTP-S9. Despite large domain size, a benefit is acquired in π-π stacking distance for PM6:BTP-S9 blend. The smallest π-π stacking distance for PM6:BTP-S9 blend is crucial for achieving efficient charge hopping between terminals in preferred vertical direction.

Further, a matrix analysis in four macroscopic factors of $\mu_e$ from SCLC, $\Delta E_3$ from energy loss analysis, $\tau_h$ and $\tau_{exciton}$ from TAS and TRPL measurements is presented in Fig. 6c to build the correlations. It's indicated that conjugation extension, molecular asymmetry and shortening alkyl side chain could show positive effects on the electron mobility, as consistent with variation trend of the domain size. Thus BTP-S9, with above three molecular structural factors included, enables the relevant blend showing the highest electron mobility, which is also an echo with the smallest π-π stacking distance. The overall hole transfer process is determined by both $\tau_1$ and $\tau_2$, which are correlated to both

exciton dissociation and diffusion processes, respectively. From the above analysis, the hole transfer seems to be very complicated, and is ultimately controlled by variety of factors, such as the HOMO offset and morphological factors, i.e., domain size, domain purity, and molecular packing. Conjugation extension of BTP-S7 results in small HOMO offset, but fast exciton dissociation can still be achieved through compensation with good molecular packing and phase-separated morphology. Molecular asymmetry seems to be harmful for the hole transfer process, which is caused by the increased domain sizes. However, shortened alkyl side chain of BTP-S9 induces more obvious phase separation for more pure domains, thus leading to faster exciton diffusion than BTP-S8 blend (Fig. 5d). The exciton lifetime of 0.83 ns for BTP-S9 is long enough to allow efficient exciton diffusion from bulk to D:A interface for efficient charge separation with domain size of 53 nm. More discussion can be found in Supplementary Note 1. Besides, all four NFAs own exciton lifetime of around 0.9 ns in neat acceptor films, thus providing enough tolerance for exciton dissociation even with large domain sizes. These results validate the importance of morphological factors for charge transfer, and explain the possible mechanism underlying the fast charge transfer with small HOMO offsets. For the $\Delta E_3$, the HOMO offset of D:A blend seems still play the dominant role, regardless of the morphological factors, in which PM6:BTP-S7-based device shows the smallest HOMO offset and also the lowest nonradiative loss. So, under the premise of long exciton lifetime, which provides enough tolerance of avoiding severe charge recombination, the BTP-S9, with three molecular structural factors included, performs the best by realizing stronger terminal packing to obtain higher electron mobility and forming more obvious phase

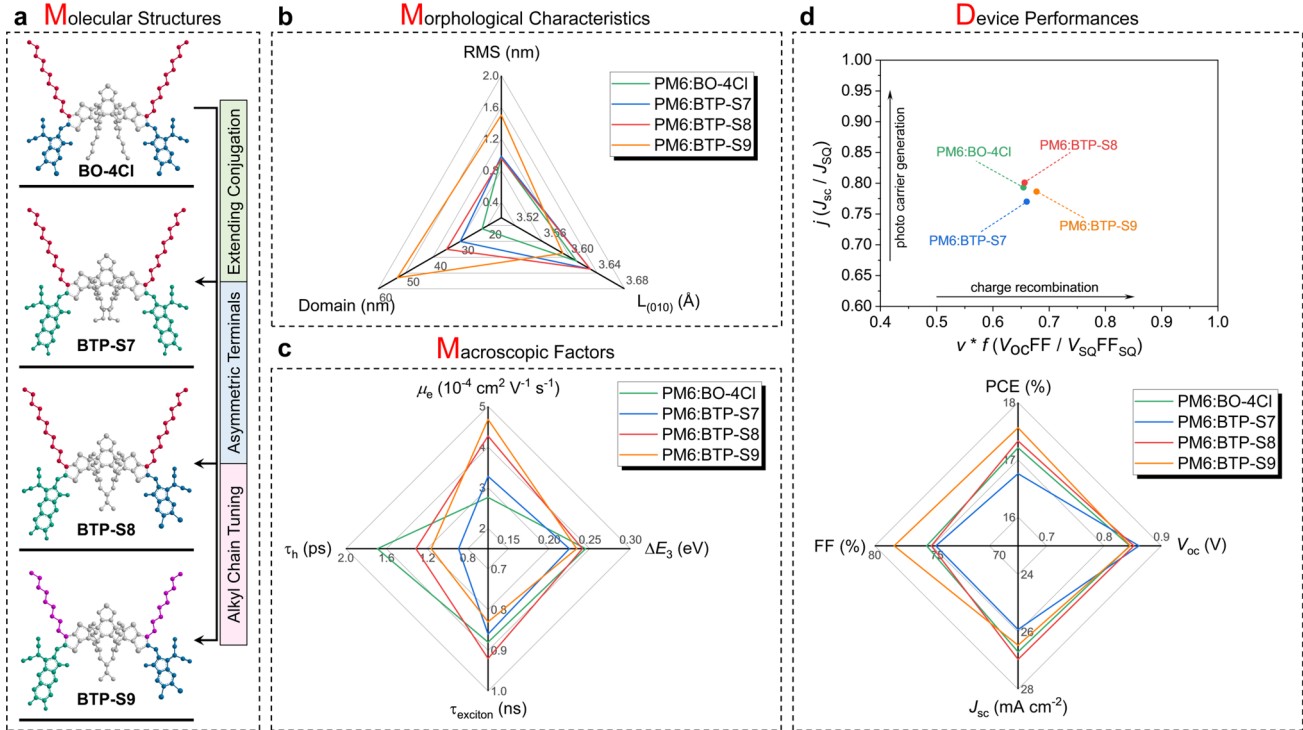

**Fig. 6 Correlation analysis. a** Molecular structures diagram of BO-4Cl, BTP-S7, BTP-S8, and BTP-S9. **b** Morphological characteristics matrix of RMS, π–π stacking distance and domain size. **c** Macroscopic factors matrix of $\mu_e$, $\Delta E_3$, $\tau_{exciton}$ (exciton lifetime for pure acceptors) and $\tau_h$ (total average hole transfer time). **d** Device performances comparison among four OPVs.

separation to maintain fast exciton diffusion for efficient hole transfer.

Finally, correlation between the macroscopic properties and device performances is depicted (Fig. 6d). Photovoltaic parameters comparison shows that $V_{oc}$ is intimately related with the $\Delta E_3$, and the direct reciprocal relation between them confirms that $V_{oc}$ is mainly limited by nonradiative loss and the HOMO offset at D:A interface. As known, the $J_{sc}$ is more complicated, and influenced by a variety of factors, including light absorption and carrier dynamics, thus making it difficult to correlate with an individual macroscopic factor. With fast charge separation and high electron mobility maintained, efficient photo-carrier generation is all presented in four PM6:NFA blends. As for FF, it shows the similar tendency as those observed in morphological characteristics. The largest phase separation of PM6:BTP-S9 blend leads to the highest electron mobility, the least nonideal charge recombination, and thereby the highest FF. We notice that though the two-fold increase in electron mobility from PM6:BO-4Cl to PM6:BTP-S9 devices, a small increase in FF from 75.83 to 78.44% is observed, indicating weak correlation between electron mobility and FF. Such weak correlation with a solid trend has been demonstrated in the previous work that the small improvement in FF requires significant increase in charge mobility when the FF is already very high[46]. As shown in Fig. 6d (upper), with values of SQ limits as the references (SQ limits are the calculated highest ideal values for single p-n junction solar cells and can be found in Supplementary Fig. 20), the three photovoltaic parameters, i.e., $J_{sc}$, $V_{oc}$, and FF are compared to their corresponding SQ limit values to generate the $j$ ($J_{sc}/J_{SQ}$), $v$ ($V_{oc}/V_{SQ}$), and $f$ (FF/FF$_{SQ}$), and these further can be divided into two categories of photo-carrier generation management representing with $j$ and charge recombination management representing with $v \times f$[47]. The first term $j$ quantifies property of utilizing photon for generating current, while the latter one $v \times f$ numerically illustrates the carrier recombination behaviors in

the whole working regions, i.e., from 0 V to $V_{oc}$. With similar $J_{sc}$ values and band gaps, these OPVs demonstrate semblable situations of photo-carrier generation management except PM6:BTP-S7. Whereas PM6:BTP-S9-based OPV shows better carrier recombination property than other three OPVs, due to the concerted benefits from $V_{oc}$ and FF as analyzed above. These results determine PM6:BTP-S9 as the best-performance system among the four NFAs, stressing that the molecular asymmetry might be one of the options for well balancing parameters trade-off. A further step in efficiency is possible if carrier generation and recombination can be better manipulated. As results, the above analysis illustrates the correlations among molecular structures, morphological characteristics, macroscopic factors, and device performances, and has great implication for developing better-performing OPVs.

## Discussion
In conclusion, we performed a multidimensional study based on state-of-the-art NFA systems to build correlations among molecular structures, morphological characteristics, macroscopic properties, and device performances. Four NFAs involved in three molecular structure features of extending conjugation, asymmetric terminals, and alkyl chain length tuning were selected or synthesized. After correlation analysis, it was found that extending conjugation at the terminals of NFAs could have a positive effect on boosting charge separation and lowering nonradiative loss, thus beneficial for a high open-circuit voltage. It was also indicated that asymmetric terminals could better balance the three device parameters for a high efficiency. It should be noticed that the length of alkyl side chain affects the molecular packing and electron mobility. Shortening the alkyl side chain could enhance the molecular packing strength and decrease the π–π distance, thus enhancing the electron mobility, finally leading to a higher FF. As a result, with the most-acquired benefits from molecular structure and macroscopic factors, the OPV based on

the designed asymmetric acceptor BTP-S9 with shorter alkyl side chain achieved the optimal PCE of 17.56% with a high FF of 78.44%. The correlations revealed in this work provide a guideline for optimizing specific factors via molecular design, thus pave the way toward developing better photoactive materials for higher device performances.

## Methods

**Materials**. PM6 and BO-4Cl were purchased from Solarmer Materials Inc. The detailed synthesis procedures of BTP-S7, BTP-S8, and BTP-S9 are described in Supplementary Methods.

**Device fabrication and measurement**. Organic photovoltaics were fabricated on glass substrates commercially pre-coated with a layer of ITO with the conventional structure of ITO/PEDOT:PSS/Active Layer/PFN-Br/Ag. Prior to fabrication, the substrates were cleaned using detergent, deionized water, acetone and isopropanol consecutively for 15 min in each step, and then treated in an ultraviolet ozone generator for 15 min before being spin coated at 4500 rpm with a layer of 20 nm thick PEDOT:PSS (Baytron P AI4083). After baking the PEDOT:PSS layer in air at 170 °C for 20 min, the substrates were transferred to a glovebox. Then the active layer was spin coated from chloroform solution (for PM6:BO-4Cl, 17.6 mg/mL in total, D:A = 1:1.2, 0.25% DIO; for PM6:BTP-S7, 16 mg/mL in total, D:A = 1:1, 0.5% CN; for PM6:BTP-S8, 16 mg/mL in total, D:A = 1:1, 0.5% CN; for PM6:BTP-S9, 17.6 mg/mL in total, D:A = 1:1.2, 0.25% DIO) at relevant speed (2500 rpm for PM6:BO-4Cl, PM6:BTP-S7 and PM6:BTP-S8; 2800 rpm for PM6:BTP-S9) for 30 s to form the active layers. Notably, the PM6:BTP-S9 solution should be heated at 60 °C for 1 h before spin coating. Then an extra preannealing at 80 °C (PM6: BTP-S7 and PM6:BTP-S9) or 100 °C (PM6:BO-4Cl and PM6:BTP-S8) for 10 min was performed. Then a 5 nm thick PFN-Br film was deposited as the cathode buffer layer by the spin coating of a solution of 0.5 mg/mL PFN-Br in methanol. Finally, the Ag (100 nm) electrode was deposited by thermal evaporation to complete the device with an active area of 9.25 mm$^2$, as defined by the overlapping area of ITO and Ag. A mask with an area of 5.979 mm$^2$ (certified by National Institute of Metrology, China) was used to measure the efficiencies. The J–V measurement was performed via the solar simulator (SS-F5-3A, Enlitech) along with AM 1.5 G spectra at 100 mV/cm$^2$, that was calibrated by the certified standard silicon solar cell (SRC-2020, Enlitech) with KG-2 filter. Devices were tested in N$_2$-filled glovebox. The scan direction is −0.2 to 1.2 V, with a scan step of 0.01 V and dwell time is 1 ms. The EQE data were obtained by using the solar-cell spectral-response measurement system (RE-R, Enlitech).

**Reporting summary**. Further information on research design is available in the Nature Research Reporting Summary linked to this article.

## Data availability

The data that support the plots within this paper and other finding of this study are available from the corresponding authors upon reasonable request.

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

## Acknowledgements

This work is supported by the National Natural Science Foundation of China (Nos. 21734008, 21875216, 51803178, 61721005), National Key Research and Development Program of China (No. 2019YFA0705900), and the China Postdoctoral Science Foundation Funded Project (Nos. 2020M671715, 2017M621907, 2019T120501). L. Zuo acknowledges the research start-up fund from Zhejiang University. N.Y. and F.Z. acknowledge the financial support from Swedish Government Strategic Research Area in Material Science on Functional Materials at Linköping University (Faculty Grant SFO-Mat-LiU No 200900971), Swedish Research Council (2017-04123), and China Scholarship Council (CSC) (No. 201708370115). X.L. and X.X. acknowledge the financial support from Research Grant Council of Hong Kong (General Research Fund No. 14303519) and CUHK Direct Grant (No. 4053415). Z.C. and H.Z. acknowledge the financial support from National Key Research and Development Program of China (No. 2017YFA0207700).

## Author contributions

S.L., L. Zuo, and H.C. conceived the idea. S.L. synthesized the nonfullerene acceptors and performed the related characterizations. L. Zhan fabricated the OPV devices and performed the related measurements. N.Y. and F.Z. carried out the energy loss test. X.X. and X.L. did the GIWAXS/GISAXS measurements. Z.C. and H.Z. carried out TRPL and TAS characterizations. W.Y. and C.H. assisted in SQ limit and DFT calculations. M.S. participated in data analysis and paper preparation. H.C. supervised the project. The manuscript was mainly written by S.L., L. Zuo, and H.C., and all authors commented on the manuscript. S.L. and L. Zhan contributed equally to this work.

## Competing interests

The authors declare no competing interests.
