## [Peer Review File · Nature Communications]

REVIEWER COMMENTS

Reviewer #1 (Remarks to the Author):

The recent years have witnessed significant progress in organic photovoltaics (OPV). Particularly, the molecular structure design and synthesis, morphology control and characterization techniques, and the analysis on the macroscopic factors have become more and more sophisticated. However, the ultimate question on what is the optimal molecular structure for best device performance remains unclear. To get over this, the fundamental understanding on the structure-performance relationship is urgently required. Considering the morphology and macroscopic factors are two factors bridging the molecular structure and the photovoltaic performance, it is necessary to correlate the molecular structure to the device performance by considering the subsequent effect of morphology and macroscopic factors. However, this topic is less explored.

In this work, Chen et al. tried to unveil the relationship among molecular structure, morphological characteristic, macroscopic properties and the photovoltaic performance of organic photovoltaics (OPVs). To my knowledge, this is the first try to unveil such a systematic puzzle in the field of OPV. They have performed a multidimensional study based on four state-of-the-art electron acceptors with three different molecular structural factors of conjugation extension, asymmetric terminals and alkyl chain tuning. The morphology of subsequent BHJ film based on the four non-fullerene acceptors (NFAs) is comprehensively studied via GIWAXS, GSAXS, AFM etc. Further, the macroscopic properties, e.g. carrier mobility, exciton lifetime, charge transfer, charge recombination, etc. are studied and correlated with the molecular structure and morphological factors. Finally, the above observations are linked to the photovoltaic performance. The establishment of a reliable correlation among molecular structures, morphological characteristics, macroscopic factors and device performances would in no doubt shed light on the future development of OPV materials. Further, the logic of this work, i.e. systematically unveiling the correlation from molecular structure, morphology, macroscopic properties and device performance, should be recommended in the field of OPV research. Besides, benefitted from multiple molecular design strategies, the designed NFA molecules in this work exhibit some unique features, e.g. the lowest energetic disorder in highly efficient OPVs, long exciton lifetime in the time scale of ns, fast charge separation under small driving force and the outstanding efficiencies for asymmetric acceptors. Therefore, this work is very timely and the results are important for researchers to design electron acceptors for high performance OPVs. I am happy to recommend this work to publish in Nature Communications after the following issues are addressed.

- 1) Urbach energy of the pristine NFA films should also be analyzed and compared with that of the photovoltaic device.
- 2) In Fig. 3e, more prominent data plotting method should be used to stress the difference of VOC loss of the four NFAs.
- 3) Small domain sizes are normally beneficial for charge separation, while efficient hole transfer can still happen in PM6:BTP-S9-based blend with large domain sizes of 53 nm. More discussion should be provided to clear it.
- 4) In Fig. 1 e-g, "(a.u.)" is not necessary for the Y-axis and should be removed.
- 5) In Fig. 4a, "nm" unit should be included after RMS values.
- 6) In Fig. 5c, the title of Y-axis should also be changed into "Normalized intensity" for universal standards.

Reviewer #2 (Remarks to the Author):

In this manuscript, Li et al. designed and synthesized three non-fullerene acceptors (NFAs), BTP-S7, BTP-S8 and BTP-S9. Together with BO-4Cl, a comprehensive study based on four NFAs, is performed. Finally, BTP-S9 with a combination of the asymmetric terminals and shortened alkyl side chain exhibited the best photovoltaic properties. In details, PM6:BTP-S9-based organic solar cells (OSCs) showed the high efficiency of 17.56% with a high fill factor of 78.44%. Overall, this work provides some interesting results, which will help to better design high-performance NFAs. However, some issues should be addressed before publishing.

- (1) Although the authors focus on studying the structure-performance relationships of molecular structures, morphological characteristics, macroscopic properties and device performances, it still

remains unclear. The BTP-S9 exhibited better photovoltaic performance than BO-4Cl, BTP-S7, BTP-S8. Actually, the molecular packing can better reflect the main differences. It would be more useful if the authors can grow the single crystals of these NFAs and compare them in details, which can help to further understand the structure-performance relationships.

(2) In this study, the authors reported an efficiency of 17.56% for BTP-S9-based OSC, representing the highest efficiency reported to date for binary OSCs based on asymmetric electron acceptors. This efficiency should be verified by a third-party certification.

(3) Figure 1f shows the absorption spectra of neat NFA films. It was found that BTP-S7 and BTP-S8 demonstrated overlapped and negligible tail state absorption edges, while BO-4Cl and BTP-S9 showed overlapped and abundant tail state absorption edges. In fact, BTP-S8 and BTP-S9 possess nearly the same chemical structures, except the minor differences in the alkyl side chain length. Why the absorption profiles differ so significantly? Figure 4c shows the GIWAXS profiles of BO-4Cl, BTP-S7, BTP-S8 and BTP-S9 neat films. BTP-S8 and BTP-S9 exhibited no difference regarding the molecular packing, orientation and crystallinity. Based on the results, it seems that the absorption difference is not from the molecular packing.

(4) In lines 296 and 297, the authors mentioned that the more obvious phase separation led to higher crystallinity for PM6:BTP-S9 blend, relative to PM6:BTP-S8 blend. The phase separation has no direct correlations with the crystallinity.

(5) Figure 4e shows the domain sizes of pure acceptor phases and PM6:BTP-S9 blends have larger domain size of 53 nm among the four PM6:NFA blends. It is known the exciton diffusion length is typically less than 15 nm. So the large phase separation is not favorable for exciton dissociation and charge transport. However, BTP-S9-based OSCs showed the highest efficiency and fill factor. The authors should explain the reason.

(6) The electron mobilities of neat NFAs should be provided.

(7) The statistical values of J_{sc} , V_{oc} and FF should be provided as well.

(8) What is the annealing temperature used for thermal stability test?

(9) Why the error so large for electron mobilities shown in Figure 2d?

Reviewer #3 (Remarks to the Author):

This is an interesting paper on an important research area. An interesting series of acceptors are studied, and clear trend in device performance is observed, with overall impressive performance. A strong range of experimental approaches are used to characterise the trend in device performance. Overall this is an interesting and informative paper. My problem with this study is that for many measurements, the differences between the different acceptors are very small, and to me appear likely to be either insignificant or within experimental error. As such, there are several cases where the authors report clear correlations between device parameters (V_{oc} etc) and measurements (acceptor LUMO) which to me look unconvincing. As some examples of this: On line 172, the authors indicate the trend in V_{oc} matches the trend in acceptor energetics. However the LUMO level of these acceptors only shifts by 10-20mV in figure 1, which I guess is close to the resolution of these experiments

Similarly the alpha values obtained from figure 2e, the Urbach tail energies from figure 3a-c and the tau 1 values from figure 5d all look almost invariant to me - I am not convinced the differences are significant / reproducible.

Other data does show clear trends (for example the electron mobilities, AFM images and EL intensities).

The manuscript would be greatly improved if the authors could carefully consider which measurements clearly show significant and reproducible trends and which not, and also which trends are large enough to explain differences in device performance.

If this is done well, then I think the manuscript would become appropriate for publication in Nat. Comm.

Point-by-Point Response Letter

Reviewer #1:

Comments:

The recent years have witnessed significant progress in organic photovoltaics (OPV). Particularly, the molecular structure design and synthesis, morphology control and characterization techniques, and the analysis on the macroscopic factors have become more and more sophisticated. However, the ultimate question on what is the optimal molecular structure for best device performance remains unclear. To get over this, the fundamental understanding on the structure-performance relationship is urgently required. Considering the morphology and macroscopic factors are two factors bridging the molecular structure and the photovoltaic performance, it is necessary to correlate the molecular structure to the device performance by considering the subsequent effect of morphology and macroscopic factors. However, this topic is less explored.

In this work, Chen et al. tried to unveil the relationship among molecular structure, morphological characteristic, macroscopic properties and the photovoltaic performance of organic photovoltaics (OPVs). To my knowledge, this is the first try to unveil such a systematic puzzle in the field of OPV. They have performed a multidimensional study based on four state-of-the-art electron acceptors with three different molecular structural factors of conjugation extension, asymmetric terminals and alkyl chain tuning. The morphology of subsequent BHJ film based on the four non-fullerene acceptors (NFAs) is comprehensively studied via GIWAXS, GSAXS, AFM etc. Further, the macroscopic properties, e.g. carrier mobility, exciton lifetime, charge transfer, charge recombination, etc. are studied and correlated with the molecular structure and morphological factors. Finally, the above observations are linked to the photovoltaic performance. The establishment of a reliable correlation among molecular structures, morphological characteristics, macroscopic factors and device performances would in no doubt shed light on the future development of OPV materials. Further, the logic of this work, i.e. systematically unveiling the correlation from molecular structure, morphology, macroscopic properties and device performance, should be recommended in the field of OPV research. Besides, benefitted from multiple molecular design strategies, the designed NFA molecules in this work exhibit some unique features, e.g. the lowest energetic disorder in highly efficient OPVs, long exciton lifetime in the time scale of ns, fast charge separation under small driving force and the outstanding efficiencies for asymmetric acceptors. Therefore, this work is very timely and the results are important for researchers to design electron acceptors for high performance OPVs. I am happy to recommend this work to publish in Nature Communications after the following issues are addressed.

1) Urbach energy of the pristine NFA films should also be analyzed and compared with that of the photovoltaic device.

Response: Thanks for the suggestion. As suggested, we have calculated the Urbach energy of the pristine NFA films. As shown in **Figure R1**, we use the absorption coefficient instead of absorbance to estimate the Urbach energy. The NFA films exhibit decreased Urbach energy of 0.193, 0.184, 0.174 and 0.165 eV for BTP-S7, BO-4Cl, BTP-S8 and BTP-S9, respectively, indicating more and more ordered molecular packing (**Figure R1b**). For organic photovoltaic (OPV) devices, we used high-resolution EQE to calculate the Urbach energy (**Fig. 3b-c** in the previous manuscript). As contrasts, the Urbach energy values of the photovoltaic devices get much smaller in the range of 0.02 - 0.021 eV, and negligible differences are observed among the different NFA based OPVs. The Urbach energy differences between the neat acceptor films and photovoltaic devices might be originated from extrinsic factors, such as film thickness and device structure. **Fig. 3b-c** have been removed to **Supplementary Fig. 14** in the revised version.

Figure R1. a) absorption coefficient and b) Urbach energy plot of pristine BO-4Cl, BTP-S7, BTP-S8 and BTP-S9 film.

For convenience, we have replaced the **Fig. 1f** with the **Figure R1a** in the revised manuscript. **Figure R1b** together with **Fig. 3b-c** in the previous manuscript have been regrouped as the new **Supplementary Fig. 14** (see **Figure R2**).

Figure R2. Incorporated as **Supplementary Fig. 14** in the revised manuscript. | **a** The calculation of Urbach energy (E_U) for BO-4Cl, BTP-S7, BTP-S8 and BTP-S9 neat films. **b** The calculation of Urbach energy (E_U) for PM6:BO-4Cl and PM6:BTP-S7-based OPVs. **c** The calculation of Urbach energy (E_U) for PM6:BTP-S8 and PM6:BTP-S9-based OPVs.

Corresponding change has been added in Page 14 in the revised manuscript as follows:

“The Urbach energy values of OPV devices based on the four NFAs exhibit negligible differences, but get significantly decreased compared to those of pristine NFA films as shown in **Supplementary Fig. 14**.”

2) In Fig. 3e, more prominent data plotting method should be used to stress the difference of VOC loss of the four NFAs.

Response: Thanks for the advice. We have replotted **Fig. 3e** with a new way as shown in **Figure R3**, which makes it convenient for readers to compare the differences in voltage losses of these four OPVs.

Figure R3. Comparison of radiative and non-radiative energy losses.

3) Small domain sizes are normally beneficial for charge separation, while efficient hole transfer can still happen in PM6:BTP-S9-based blend with large domain sizes of 53 nm. More discussion should be provided to clear it.

Response: Thanks for the comment. The hole transfer contains two processes: 1) exciton diffusion towards donor/acceptor (D/A) interfaces; 2) charge separation at the D/A interfaces. The domain sizes mainly affect the exciton diffusion process, so the allowed domain sizes for OPVs rely on the exciton diffusion length. Normally, the exciton diffusion length should be larger than the half of domain sizes. The exciton diffusion length (L_D) can be calculated according to the formula of $L_D = \sqrt{D\tau}$, wherein D is the exciton diffusion constant and τ is the exciton lifetime. For Y-series non-fullerene acceptors, the exciton diffusion constant could be larger than $10^{-2} \text{ cm}^2 \text{ s}^{-1}$ (*J. Am. Chem. Soc.* 2019, 141, 6922-6929). For BTP-S9, its exciton lifetime is as long as 0.83 ns. Thus, the exciton diffusion length is at least larger than 29 nm, in other words, the allowed domain sizes can be around 58 nm. So, the pure acceptor domain size of 53 nm is still in the reasonable range for efficient hole transfer in PM6:BTP-S9 blend.

The corresponding changes have been incorporated into the manuscript in page 23 as below:

“The exciton lifetime of 0.83 ns for BTP-S9 is long enough to allow efficient exciton diffusion from bulk to D:A interface for efficient charge separation with domain size of 53 nm. More discussion can be found in **Supplementary Note 1**.”

Supplementary Note 1 is added in the **Supplementary Information** as follows:

“**Supplementary Note 1**

According to the formula of $L_D = \sqrt{D\tau}$, wherein L_D is the exciton diffusion length, D

is the exciton diffusion constant (for efficient non-fullerene acceptors, e.g. Y6, it can be in the order of magnitude of $10^{-2} \text{ cm}^2 \text{ s}^{-1}$, *J. Am. Chem. Soc.* 2019, *141*, 6922-6929), and τ is the exciton lifetime, the long exciton lifetime of 0.83 ns for BTP-S9 can allow an exciton diffusion length larger than 29 nm, leading to a tolerant pure acceptor domain size of 58 nm. So, the 53 nm pure acceptor domain size in PM6:BTP-S9 blend is still in the reasonable range for maintaining efficient hole transfer. So it is for BO-4Cl, BTP-S7 and BTP-S8 with even longer exciton lifetimes. Similar results were also observed in other reported works that a long carrier diffusion length of 41 nm was detected in PM6:Y6-based OPVs with an active layer thickness of 100 nm (*Adv. Energy Mater.* 2021, 2100804).”

4) In Fig. 1 e-g, “(a.u.)” is not necessary for the Y-axis and should be removed.

Response: Thanks for the advice. We have removed “(a.u.)” from Fig. 1e-g, and the updated Fig. 1e-g is presented as below:

Figure R4. Revised Figure 1e-g. Molecular structure, optical and electrochemical properties. e Normalized absorption spectra of BO-4Cl, BTP-S7, BTP-S8 and BTP-S9 in chloroform solution. f Absorption coefficients of BO-4Cl, BTP-S7, BTP-S8 and BTP-S9 thin films. g Normalized absorption spectra of TIC and TNC thin films.

5) In Fig. 4a, “nm” unit should be included after RMS values.

Response: Thanks for the suggestion. We have added “nm” unit after RMS values in Fig. 4a, and the updated Fig. 4a can be found below:

Figure R5. Revised Fig. 4. Morphological characterization for films. a AFM height images of PM6:BO-4Cl, PM6:BTP-S7, PM6:BTP-S8 and PM6:BTP-S9 blends.

6) In Fig. 5c, the title of Y-axis should also be changed into “Normalized intensity” for universal standards.

Response: Thanks for the advice. We have changed the title of Y-axis as “Normalized intensity” in Fig. 5c, and the updated Fig. 5c is displayed as below:

Figure R6. Revised Fig. 5. Hole transfer dynamics. c Hole transfer kinetics in four blend films.

Reviewer #2:

Comments:

In this manuscript, Li et al. designed and synthesized three non-fullerene acceptors (NFAs), BTP-S7, BTP-S8 and BTP-S9. Together with BO-4Cl, a comprehensive study based on four NFAs, is performed. Finally, BTP-S9 with a combination of the asymmetric terminals and shortened alkyl side chain exhibited the best photovoltaic properties. In details, PM6:BTP-S9-based organic solar cells (OSCs) showed the high efficiency of 17.56% with a high fill factor of 78.44%. Overall, this work provides some interesting results, which will help to better design high-performance NFAs. However, some issues should be addressed before publishing.

1) Although the authors focus on studying the structure-performance relationships of molecular structures, morphological characteristics, macroscopic properties and device performances, it still remains unclear. The BTP-S9 exhibited better photovoltaic performance than BO-4Cl, BTP-S7, BTP-S8. Actually, the molecular packing can better reflect the main differences. It would be more useful if the authors can grow the single crystals of these NFAs and compare them in details, which can help to further understand the the structure-performance relationships.

Response: Thanks for the advice. As suggested, we have tried to grow single crystals of these molecules. But we failed to grow perfect single crystals to derive the detailed molecular packing data. Optical microscopy images of cultivated crystals for BTP-S7 and BTP-S8 are shown below, but these crystals still can't reach the requirements for crystalline structure analysis. For BTP-S9, the poor solubility leads to fast precipitation, and this makes it difficult to get good crystals, for single crystal cultivation requires a slow growth.

Figure R7. Pictures of grown crystals of BPT-S7 and BTP-S8.

Notably, the previous work demonstrates that the single crystal packing feature can be inherited to the thin film (*Nat. Commun.* 2020, 11, 3943), and the main features of molecular packing from GIWAXS measurement, i.e. π - π stacking and lamellar stacking, are in good agreement with the single crystal information (see **Figure R8**).

Results from reference of "Nat. Commun. 2020, 11, 3943"

Figure R8. Single crystal packing of Y6 and GIWAXS patterns of Y6 based NFA films.

Recently, Tobin J. Marks et al. reported the single crystal structure of BT-BO-L4F, which has the same structure with the BTP-S7 in our work (see **Figure R9**). Their work also proved that molecular packing modes of Y-series single crystal could be retained in the thin film, for packing features from the GIWAXS measurements are in good agreement with the single crystal data. Therefore, the GIWAXS can also tell the main features of molecular packing in single crystals for Y6 and its derivatives, and the corresponding GIWAXS results have been analyzed in section of "Film Morphology".

Results from reference of "J. Am. Chem. Soc. 2021, 143, 6123-6139"

Figure R9. Single crystal packing of BT-BO-L4F (or BTP-S7 in our work) and GIWAXS patterns of the corresponding films. Reprinted with permission from Li, G. et al. Systematic merging of nonfullerene acceptor π -extension and tetrafluorination strategies affords polymer solar cells with >16% efficiency. *J. Am. Chem. Soc.* **143**, 6123-6139 (2021). Copyright 2021 American Chemical Society.

An illustration has been added to the manuscript in page 17 as below:

“Above distinct features of molecular packing from the acceptors in the blends might also reveal the molecular packing information in the single crystals of relevant NFAs, as demonstrated in previous works.^{39,44}”

The above-mentioned works are cited as ^{39,44}

39. Zhang, G. *et al.* Delocalization of exciton and electron wavefunction in nonfullerene acceptor molecules enables efficient organic solar cells. *Nat. Commun.* **11**, 3943 (2020).

44. Li, G. *et al.* Systematic merging of nonfullerene acceptor π -extension and tetrafluorination strategies affords polymer solar cells with >16% efficiency. *J. Am. Chem. Soc.* **143**, 6123-6139 (2021).

2) In this study, the authors reported an efficiency of 17.56% for BTP-S9-based OSC, representing the highest efficiency reported to date for binary OSCs based on asymmetric electron acceptors. This efficiency should be verified by a third-party certification.

Response: Thanks for the comment. As requested, we have sent the best-performing PM6:BTP-S9-based OPVs to “National Institute of Metrology (NIM), China” for certification, and a certified efficiency of 17.4% was achieved, with small difference to the best efficiency of 17.56% measured in our lab. The efficiency certification report has been provided below. This figure is incorporated as **Supplementary Fig. 6** in the revised manuscript.

Figure R10. Efficiency certification report for OPV based on PM6:BTP-S9 blend from National Institute of Metrology (NIM), China.

A sentence has been added in the text of page 9 as follows:

“A certified efficiency of 17.4% was achieved for PM6:BTP-S9-based OPVs from National Institute of Metrology (NIM), China (**Supplementary Fig. 6**).”

3) Figure 1f shows the absorption spectra of neat NFA films. It was found that BTP-S7 and BTP-S8 demonstrated overlapped and negligible tail state absorption edges, while BO-4Cl and BTP-S9 showed overlapped and abundant tail state absorption edges. In fact, BTP-S8 and BTP-S9 possess nearly the same chemical structures, except the minor differences in the alkyl side chain length. Why the absorption profiles differ so significantly? Figure 4c shows the GIWAXS profiles of BO-4Cl, BTP-S7, BTP-S8 and BTP-S9 neat films. BTP-S8 and BTP-S9 exhibited no difference regarding the molecular packing, orientation and crystallinity. Based on the results, it seems that the absorption difference is not from the molecular packing.

Response: Thanks for the insightful comments. Considering that the absorbance of NFA films can be influenced by the extrinsic factors, such as thickness of film, judging the band tail state absorbing properties via absorbance is quite misleading. Instead, to fairly compare the absorption properties, we used the absorbing coefficient-wavelength plot, as shown in **Figure R11** below.

Figure R11. Incorporated as **Fig. 1f** in the revised manuscript. **Molecular structure, optical and electrochemical properties.** **f** Absorption coefficients of BO-4Cl, BTP-S7, BTP-S8 and BTP-S9 thin films.

As shown in **Figure R11**, BTP-S8 and BTP-S9 show similar situation in tail state absorption edges, which is consistent with the results observed in the GIWAXS characterization.

The discussion about thin film absorption has been updated in page 8 of the manuscript as below:

“In thin films, these four NFAs exhibit similar absorption profiles. The BO-4Cl, BTP-S7 and BTP-S8 showed similar absorption coefficients of $\sim 1.3 \times 10^5 \text{ cm}^{-1}$ at the highest peaks, while a higher coefficient of $\sim 1.5 \times 10^5 \text{ cm}^{-1}$ was observed for BTP-S9 at the highest peak. Notably, the similar absorption profiles provide a unique platform to study the individual role of the molecular structure variations on carrier dynamics without entangling with the optical absorbing effect.”

4) In lines 296 and 297, the authors mentioned that the more obvious phase separation led to higher crystallinity for PM6:BTP-S9 blend, relative to PM6:BTP-S8 blend. The phase separation has no direct correlations with the crystallinity.

Response: Thanks for the comment. We agree with the reviewer that the phase separation has no direct correlations with the crystallinity, so we have removed the questionable sentence of “The more obvious phase separation led to higher crystallinity for PM6:BTP-S9 blend, relative to PM6:BTP-S8 blend, which was also confirmed below” from the manuscript in page 16.

(5) Figure 4e shows the domain sizes of pure acceptor phases and PM6:BTP-S9 blends have larger domain size of 53 nm among the four PM6:NFA blends. It is known the exciton diffusion length is typically less than 15 nm. So the large phase separation is not favorable for excitation dissociation and charge transport. However, BTP-S9-based OSCs showed the highest efficiency and fill factor. The authors should explain the reason.

Response: Thanks for the question. The domain sizes mainly affect the exciton diffusion process before exciton dissociation at the D/A interfaces. To maintain efficient charge separation, normally the required exciton diffusion length should be larger than half of the domain sizes. For fullerene-derivatives, limited by the exciton diffusion coefficient ($10^{-4} \text{ cm}^2 \text{ s}^{-1}$) and exciton lifetime, the exciton diffusion length is typically less than 15 nm (*J. Am. Chem. Soc.* 2019, 141, 6922-6929). However, for non-fullerene acceptors, especially for Y-series molecules, it's possible to realize a larger exciton diffusion length. The exciton diffusion length is decided by the formula of $L_D = \sqrt{D\tau}$, wherein L_D is the exciton diffusion length, D is the exciton diffusion constant (for Y6-type non-fullerene acceptors, it can be larger than $10^{-2} \text{ cm}^2 \text{ s}^{-1}$, *J. Am. Chem. Soc.* 2019, 141, 6922-6929), and τ is the exciton lifetime. In our cases, the BTP-S9 possesses a long exciton lifetime of 0.83 ns, thus the calculated exciton diffusion length can be larger than 29 nm. So the tolerant pure acceptor domain size can be larger than 58 nm. Therefore, the domain size of 53 nm in PM6:BTP-S9 is still acceptable for maintaining efficient charge separation. Besides, the large domain size of 53 nm in PM6:BTP-S9 leads to purer domain and higher electron mobility, both of which are beneficial for achieving a higher fill factor, thus a higher efficiency.

The corresponding changes have been added into the manuscript in page 23 as below:

“The exciton lifetime of 0.83 ns for BTP-S9 is long enough to allow efficient exciton diffusion from bulk to D:A interface for efficient charge separation with domain size of 53 nm. More discussion can be found in **Supplementary Note 1**.”

Supplementary Note 1 is added in the **Supplementary Information** as follows:

“**Supplementary Note 1**

According to the formula of $L_D = \sqrt{D\tau}$, wherein L_D is the exciton diffusion length, D is the exciton diffusion constant (for efficient non-fullerene acceptors, e.g. Y6, it can be in the order of magnitude of $10^{-2} \text{ cm}^2 \text{ s}^{-1}$, *J. Am. Chem. Soc.* 2019, 141, 6922-6929), and τ is the exciton lifetime, the long exciton lifetime of 0.83 ns for BTP-S9 can allow an exciton diffusion length larger than 29 nm, leading to a tolerant pure acceptor domain size of 58 nm. So, the 53 nm pure acceptor domain size in PM6:BTP-S9 blend is still

in the reasonable range for maintaining efficient hole transfer. So it is for BO-4Cl, BTP-S7 and BTP-S8 with even longer exciton lifetimes. Similar results were also observed in other reported works that a long carrier diffusion length of 41 nm was detected in PM6:Y6-based OPVs with an active layer thickness of 100 nm (*Adv. Energy Mater.* 2021, 2100804).”

(6) The electron mobilities of neat NFAs should be provided.

Response: Thanks for the advice. We have measured the electron mobilities of neat NFAs via space-charge limited current (SCLC) method. The results are provided as **Supplementary Fig. 10** as below. Due to the strong crystallinity and limited solubility, BTP-S9 shows poor film-forming ability, thus making it difficult for us to fabricate electron-only devices with enough thick films. It was found that electron mobilities of neat NFAs were in similar values of $\sim 10^{-4} \text{ cm}^2 \text{ V}^{-1} \text{ s}^{-1}$.

Figure R12. Incorporated as **Supplementary Fig. 10** in the revised manuscript. $J^{0.5}$ - V curves of the electron-only devices based on BO-4Cl, BTP-S7 and BTP-S8 neat films.

Supplementary Fig. 10 showing the results of neat acceptor electron mobilities has been added in the Supplementary Information.

The following sentence has been added to the manuscript in page 11 as below:

“For neat acceptors, they showed similar electron mobilities in the order of magnitude of $10^{-4} \text{ cm}^2 \text{ V}^{-1} \text{ s}^{-1}$ (**Supplementary Fig. 10**).”

(7) The statistical values of J_{sc} , V_{oc} and FF should be provided as well.

Response: Thanks for the advice. The statistical values of J_{sc} , V_{oc} and FF have been added in **Table 1** as follows:

Table 1 Photovoltaic parameters of the OPVs based on PM6:NFA blends.

Blend ^a	V_{oc} (V)	J_{sc} (mA cm ⁻²)	FF (%)	PCE (%) ^b	$J_{cal.}$ (mA cm ⁻²) ^c	$\mu_h \times 10^4$ (cm ² V ⁻¹ s ⁻¹) ^d	$\mu_e \times 10^4$ (cm ² V ⁻¹ s ⁻¹) ^e
PM6:BO- 4Cl	0.845 (0.846±0.002)	26.70 (26.62±0.13)	75.83 (75.20±0.53)	17.21 (17.02±0.15)	26.39	2.50±0.62	2.76±0.83
PM6:BTP- S7	0.861 (0.859±0.003)	25.92 (26.00±0.12)	75.09 (73.62±0.59)	16.76 (16.51±0.16)	25.87	2.29±0.35	3.28±0.70
PM6:BTP- S8	0.852 (0.852±0.002)	26.96 (26.66±0.36)	75.45 (74.99±1.05)	17.33 (17.05±0.11)	26.56	2.27±0.30	4.27±1.34
PM6:BTP- S9	0.846 (0.848±0.003)	26.47 (26.48±0.14)	78.44 (77.72±0.46)	17.56 (17.42±0.07)	26.42	2.40±0.20	4.68±0.20

(8) What is the annealing temperature used for thermal stability test?

Response: Thanks for the question. The annealing temperature used for thermal stability test was 80 °C, and we have added the annealing temperature on **Fig. 2c** as follows:

Figure R13. Incorporated as **Fig. 2c** in the revised manuscript. **Photovoltaic performance, charge transport and recombination. c** Light stability and thermal stability comparisons of relevant OPVs.

(9) Why the error so large for electron mobilities shown in Figure 2d?

Response: Thanks for the advice. The large error is originated from few discrete points. We have replotted **Fig. 2d** with all data listed (see Figure below). It clearly shows that most data are located around the average values. The variation of electron mobility might be originated from the morphology or diverse molecular packing of NFAs when blending with PM6.

Figure R14. Incorporated as **Fig. 2d** in the revised manuscript. **Photovoltaic performance, charge transport and recombination. d** Hole mobility and electron mobility comparisons of relevant blends.

Reviewer #3:

Comments:

This is an interesting paper on an important research area. An interesting series of acceptors are studied, and clear trend in device performance is observed, with overall impressive performance. A strong range of experimental approaches are used to characterise the trend in device performance. Overall this is an interesting and informative paper. My problem with this study is that for many measurements, the differences between the different acceptors are very small, and to me appear likely to be either insignificant or within experimental error.

Response: Thanks for the comments. We agree that in this work the variation in molecule structures, optoelectronic properties, morphology, etc. is not very large or insignificant. But considering the progress on OPV with efficiency approaching 19%, it is time to study these details of small variations, in order to delicately optimize the device performance. On the other hand, the previous research has established some well-known relationship to illustrate effects of very large variation of these parameters on device performance, and based on the previous experience, this work is carried out to refine these parameters.

As such, there are several cases where the authors report clear correlations between device parameters (V_{oc} etc) and measurements (acceptor LUMO) which to me look unconvincing. As some examples of this:

On line 172, the authors indicate the trend in V_{ox} matches the trend in acceptor energetics. However the LUMO level of these acceptors only shifts by 10-20mV in figure 1, which I guess is close to the resolution of these experiments

Response: Thanks for the nice comments. We agree there is very small shift in LUMO levels of these NFAs, which is close to the resolution of the instruments. However, variation trend of LUMO is more important in this work. We are confident on the LUMO variation trend of the four NFAs, as independently confirmed by two techniques: CV, DFT calculation.

While, the V_{oc} is actually very accurate. For a voltage source current measurement mode, the accuracy of V_{oc} is actually determined by the current, according to $\Delta V = \Delta I * R_s$. In our case, Keithley 2400 used in our lab has an accuracy of 10^{-12} A with compliance of 0.1 A, and the series resistance of our OPV is around 60~100 Ω . As a result, the sensitivity for V_{oc} is estimated to be around 10^{-10} V, and validates the authenticity of V_{oc} value derived. Besides of the extrinsic instrument factor, the intrinsic device structural factors, e.g. morphology, interface, etc. also contribute to the variation of V_{oc} , the histogram of V_{oc} is drawn and added in **Supplementary Fig. 5** (see **Figure R15**), which shows a clear variation trend with LUMOs of NFAs.

Figure R15. Incorporated as **Supplementary Fig. 5** in the revised manuscript. Voltage statistics of OPVs based on PM6:BTP-S7, PM6:BTP-S8, PM6:BTP-S9 and PM6:BO-4Cl blends.

Figure R16. J - V characteristic curves for estimating the accuracy of V_{oc} from instrument error.

To exclude the possible batch to batch variation, the statistical data of V_{oc} are also included in **Table 1** as shown below:

Table 1 Photovoltaic parameters of the OPVs based on PM6:NFA blends.

Blend ^a	V_{oc} (V)	J_{sc} (mA cm ⁻²)	FF (%)	PCE (%) ^b	$J_{cal.}$ (mA cm ⁻²) ^c	$\mu_h \times 10^4$ (cm ² V ⁻¹ s ⁻¹) ^d	$\mu_e \times 10^4$ (cm ² V ⁻¹ s ⁻¹) ^e
PM6:BO- 4Cl	0.845 (0.846±0.002)	26.70 (26.62±0.13)	75.83 (75.20±0.53)	17.21 (17.02±0.15)	26.39	2.50±0.62	2.76±0.83
PM6:BTP- S7	0.861 (0.859±0.003)	25.92 (26.00±0.12)	75.09 (73.62±0.59)	16.76 (16.51±0.16)	25.87	2.29±0.35	3.28±0.70
PM6:BTP- S8	0.852 (0.852±0.002)	26.96 (26.66±0.36)	75.45 (74.99±1.05)	17.33 (17.05±0.11)	26.56	2.27±0.30	4.27±1.34
PM6:BTP- S9	0.846 (0.848±0.003)	26.47 (26.48±0.14)	78.44 (77.72±0.46)	17.56 (17.42±0.07)	26.42	2.40±0.20	4.68±0.20

Similarly the alpha values obtained from figure 2e, the Urbach tail energies from figure 3a-c and the tau 1 values from figure 5d all look almost invariant to me - I am not convinced the differences are significant / reproducible.

Response: Thanks for the comments. As suggested, we have removed the analysis on the α values shown in **Fig. 2e** in previous manuscript, for they are indeed quite similar and approaching the limit of OPV. Thereafter, these results can only indicate that the four OPVs are all working efficiently with high FF maintained, and the charge recombination is well managed.

The discussion on α values have been simplified as below in page 12.

“The α values were calculated as 0.996 for PM6:BO-4Cl-based OPVs, 0.995 for both PM6:BTP-S7 and PM6:BTP-S8-based OPVs, and 0.999 for PM6:BTP-S9-based OPVs (**Fig. 2e**), representing similar situations again.”

As to **Fig. 3a-c** in the previous manuscript, we agree with reviewer that variations of Urbach energy among these four OPVs are very small (< 0.6 meV), which makes Urbach energy an insignificant factor to correlate with device performances. So, we have removed the correlation analysis based on the Urbach energy in page 23 (Previous correlation analysis about Urbach energy: **While the E_u in D:A blend seems to inherit the trend of pristine NFA films, which is a result of improved molecular order with longer conjugation in terminal groups. While the molecular symmetry and side chain modification seem to play a relatively minor role.**) Besides, **Fig. 3b-c** in the previous manuscript have now been moved to **Supplementary Fig. 14** (see **Figure R17**)

Figure R17. Incorporated as **Supplementary Fig. 14** in the revised manuscript. **b** The calculation of Urbach energy (E_U) for PM6:BO-4Cl and PM6:BTP-S7-based OPVs. **c** The calculation of Urbach energy for PM6:BTP-S8 and PM6:BTP-S9-based OPVs.

Since Urbach energy has been removed for correlation analysis, **Fig. 6c** has also been replotted (see **Figure R18**).

Figure R18. Incorporated as **Fig. 6** in the revised manuscript. **Correlation analysis.** **a** Molecular structures diagram of BO-4Cl, BTP-S7, BTP-S8 and BTP-S9. **b** Morphological characteristics matrix of RMS, π - π stacking distance and domain size. **c** Macroscopic factors matrix of μ_e , ΔE_3 , τ_{exciton} (exciton lifetime for pure acceptors) and τ_h (total average hole transfer time). **d** Device performances comparison among four OPVs.

The hole transfer contains two process: (1) exciton dissociation at the interfaces, representing with τ_1 ; (2) exciton diffusion assisted interfacial hole transfer process, representing with τ_2 . We agree with reviewer that τ_1 values from **Fig. 5d** are similar (indicating all four OPVs could achieve fast charge separation at the D/A interfaces),

thus, the exciton diffusion process makes a significant difference for these four OPVs due to the large changes in electron mobilities and domain sizes. So, in the correlation analysis, we have used the total average hole transfer time of τ_h , for which both τ_1 and τ_2 are considered ($\tau_h = A_1\tau_1 + A_2\tau_2$, A_1 is the ratio of τ_1 , A_2 is the ratio of τ_2), as the macroscopic factor to compare, and obvious variations could be observed for these four OPVs.

Other data does show clear trends (for example the electron mobilities, AFM images and EL intensities).

The manuscript would be greatly improved if the authors could carefully consider which measurements clear show significant and reproducible trends and which not, and also which trends are large enough to explain differences in device performance.

If this is done well, then I think the manuscript would become appropriate for publication in Nat. Comm.

Response: Thanks for the constructive advice, and we really appreciate it. After analyzing all the morphological characteristics and macroscopic factors, we find α value, Urbach energy and τ_1 value don't show obvious variations to correlate with device performances, thus the relevant parts related with the above properties have been simplified or removed in the revised manuscript (illustrated above), while properties including domain size, electron mobility, electroluminescence and exciton diffusion process show clear and large enough variations to explain the differences in device performance.

Corresponding texts, which were involved with Urbach energy previously, have been re-edited to remove uncertain conclusions originated from Urbach energy in page 8 (analysis about TIC and TNC) and page 25 ("Discussion" section). More descriptions about $J-V$ measurement conditions have been added in the section of "Device fabrication and measurement" in the manuscript, according to the requirements by "Nature research | solar cells reporting summary".

We have made the changes accordingly, and the revised manuscript has been much improved now. Hopefully, the revised manuscript is appropriate for publication in *Nat. Commun.*

REVIEWERS' COMMENTS

Reviewer #1 (Remarks to the Author):

The authors have revised the manuscript according to my concerns and I suggest it can be accepted for publication as it is.

Reviewer #2 (Remarks to the Author):

After revision, the quality of this work has been improved. I therefore recommend the acceptance of this paper in its current form.

Reviewer #3 (Remarks to the Author):

This is a detailed analysis of the performance of organic solar cells employing four different NFAs. The device performance, and particularly the fill factors, are impressive. Overall the quality of the work, and its discussion is very high. However, the focus of the paper is on a comparison of the four NFAs, which overall show very similar device performance and very similar characterisation results. The most notable difference is probably the two fold increase in electron mobility for the S9 derivative, correlated with a small increase in FF from 75.8 to 77.4. The correlation analysis in Figure 6, but to me the trends are so small, it is hard to be convinced they are significant. As such, while I think this is an solid and detailed study, I am not convinced the trends observed, and the conclusions which can be drawn, are important enough for nature comm.

Point-by-Point Response Letter

Reviewer #1:

Comments:

The authors have revised the manuscript according to my concerns and I suggest it can be accepted for publication as it is.

Response: Thank the reviewer very much for his/her positive assessments.

Reviewer #2:

Comments:

After revision, the quality of this work has been improved. I therefore recommend the acceptance of this paper in its current form.

Response: Thank the reviewer very much for his/her positive assessments.

Reviewer #3:

Comments:

This is a detailed analysis of the performance of organic solar cells employing four different NFAs. The device performance, and particularly the fill factors, are impressive. Overall the quality of the work, and its discussion is very high. However, the focus of the paper is on a comparison of the four NFAs, which overall show very similar device performance and very similar characterisation results. The most notable difference is probably the two fold increase in electron mobility for the S9 derivative, correlated with a small increase in FF from 75.8 to 77.4. The correlation analysis in Figure 6, but to me the trends are so small, it is hard to be convinced they are significant. As such, while I think this is an solid and detailed study, I am not convinced the trends observed, and the conclusions which can be drawn, are important enough for nature comm.

Response: Thank the reviewer very much for the constructive comments. We appreciate the reviewer's recognition on this work as high-quality. We agree with the reviewer that variations of devices parameter and characterization results are small. Regarding the correlation between carrier mobility and FF, it has been demonstrated that increasing the carrier mobility even by one order of magnitude will not significantly improve the FF when the FF is already very high, as shown in **Figure R1** (*Nat. Commun.* **6**, 6951 (2015)). As a result, the improvement of FF from 75.8% to 78.4% is a solid, although it is very small. Therefore, the trends observed with these small but solid variations should be convincing.

Figure R1. *J-V* curves for six different charge-carrier mobilities (Results from *Nat. Commun.* **6**, 6951 (2015)).

To avoid any possible over-toned conclusions or trends, we have lowered our tone on the conclusions or added caveats on the correlation analysis in the revised manuscript, as suggested.

In “Abstract” section, “A more significant molecular optimization is from alkyl chain tuning.” is changed into “**Another** molecular optimization is from alkyl chain tuning.”

In Page 7, the word “significantly” has been removed from the sentence of “Normally, terminal tuning will significantly affect the energy levels.”

In Page 14, the word “significantly” has been removed from the sentence of “but get significantly decreased compared to those of pristine NFA films...”.

In Page 15, the word “significantly” has been removed from the sentence of “indicating the length of the alkyl side chain could significantly affect the phase separation behavior.”

In page 22, the word “significant” and “effectively” have been removed from the sentence of “Notably, shortening alkyl side chain (C9) from BTP-S8 to BTP-S9 significantly increases RMS and domain size of the pure acceptor phase, indicating terminal packing may be effectively strengthened for BTP-S9.”

In Page 22, the word “significant” has been removed from the sentence of “a significant benefit is acquired in π - π stacking distance for PM6:BTP-S9 blend.”

In page 24, the following sentences are incorporated:

“We notice that though the two fold increase in electron mobility from PM6:BO-4Cl to PM6:BTP-S9 devices, a small increase in FF from 75.83% to 78.44% is observed, indicating weak correlation between electron mobility and FF. Such weak correlation with a solid trend has been demonstrated in the previous work that the small improvement in FF requires significant increase in charge mobility when the FF is

already very high.⁴⁶”

In page 24, the word “significantly” has been removed from the sentence of: “Whereas PM6:BTP-S9-based OPV shows significantly better carrier recombination property than other three OPVs, due to the concerted benefits from V_{oc} and FF as analyzed above.”

In “Discussion” section, “It was worth pointing out, the length of alkyl side chain had significant effects on molecular packing and electron mobility.” has been changed into “It **should be noticed that** the length of alkyl side chain **affects the** molecular packing and electron mobility.”

The above-mentioned reference has been cited as “⁴⁶” as below:

46. Würfel, U., Neher, D., Spies, A. & Albrecht, S. Impact of charge transport on current-voltage characteristics and power-conversion efficiency of organic solar cells. *Nat. Commun.* **6**, 6951 (2015).

Hopefully, this final revised version is appropriate for publication in *Nature Communications*.